# Graphical Models in Heavy-Tailed Markets

**José Vinícius de M. Cardoso, Jiaxi Ying, Daniel P. Palomar**
Department of Electronic and Computer Engineering
Department of Industrial Engineering and Decision Analytics
The Hong Kong University of Science and Technology
Clear Water Bay, Hong Kong SAR China
`{jvdmc, jx.ying}@connect.ust.hk, palomar@ust.hk`

## Abstract

Heavy-tailed statistical distributions have long been considered a more realistic statistical model for the data generating process in financial markets in comparison to their Gaussian counterpart. Nonetheless, mathematical nuisances, including nonconvexities, involved in estimating graphs in heavy-tailed settings pose a significant challenge to the practical design of algorithms for graph learning. In this work, we present graph learning estimators based on the Markov random field framework that assume a Student-$t$ data generating process. We design scalable numerical algorithms, via the alternating direction method of multipliers, to learn both connected and $k$-component graphs along with their theoretical convergence guarantees. The proposed methods outperform state-of-the-art benchmarks in an extensive series of practical experiments with publicly available data from the S&P500 index, foreign exchanges, and cryptocurrencies.

## 1 Introduction

Graph learning frameworks are often designed based on the assumption that the observed graph signals are Gaussian distributed [1–9]. While such assumption for graphical models has found great success in many practical areas, which includes brain network analysis [10], psychological networks [11], and single-cell sequencing [12], it inherently neglects scenarios where there may exist outliers or the underlying data is naturally heavy-tailed distributed. As a consequence, those methods often lack robustness and may not succeed in capturing a meaningful representation of the underlying graph [13].

Data from financial instruments are well-known examples of such scenarios where heavy-tailedness and skewedness are present [14–19]. In addition, there has been a growing interest in methods for estimating graphical models in financial markets, which hence demands the development of scalable and robust learning algorithms [20].

Perhaps one of the most prominent applications, clustering financial time-series via graph techniques has been an active research topic [20–24]. Nonetheless, current techniques rely on the assumption that the underlying graph has a tree structure, which does bring advantages due to its hierarchical clustering properties, but also have been shown to be unstable [25–27] and not suitable when the data is not Gaussian distributed [28].

Motivated by practical challenging applications in finance, such as clustering of financial instruments and network estimation, we investigate the problem of learning graph matrices whose structure follow that of a Laplacian matrix of an undirected weighted graph for which the data generating process is assumed to be Student-$t$ distributed. In particular, the main contributions of this paper are as follows:

- We propose a novel formulation for learning undirected weighted graphs under the assumption that the data generating process is Student-$t$ distributed. We solve the underlying

learning problem via a carefully designed numerical algorithm based on the alternating direction method of multipliers (ADMM), along with the establishment of its theoretical convergence guarantees. We note that the proposed algorithm can be easily extended to account for additional linear constraints on the graph weights.

- We extend the proposed framework to account for heavy-tails and $k$-component graphs simultaneously, which enables a novel method for clustering financial time-series.

- We present extensive practical results, with real-world data from the US stock market, foreign exchanges, and cryptocurrencies, that showcase clear advantages of including heavy-tail assumptions into graph learning frameworks when compared to state-of-the-art, Gaussian-based methods.

**Notation**: Matrices (vectors) are denoted by bold, italic, capital (lowercase) roman letters like $\boldsymbol{X}, \boldsymbol{x}$. Vectors are assumed to be column vectors. The $(i, j)$ element of a matrix $\boldsymbol{X} \in \mathbb{R}^{n \times p}$ is denoted as $X_{ij}$. The $i$-th element of a vector $\boldsymbol{x}$ is denoted as $x_i$. The $i$-th row of $\boldsymbol{X}$ is denoted as $\boldsymbol{x}_i \in \mathbb{R}^{p \times 1}$. Given a symmetric matrix $\boldsymbol{A}$, $\lambda_i(\boldsymbol{A})$ and $\lambda_{\max}(\boldsymbol{A})$ denote the $i$-th smallest and maximum eigenvalue of $\boldsymbol{A}$, respectively. The Moore-Penrose inverse of $\boldsymbol{A}$ is denoted as $\boldsymbol{A}^\dagger$. The Frobenius norm of a matrix $\boldsymbol{A}$ is denoted as $\|\boldsymbol{A}\|_{\mathrm{F}} = \sqrt{\mathsf{tr}\left(\boldsymbol{A}^\top \boldsymbol{A}\right)}$. The operator $\mathsf{Diag} : \mathbb{R}^p \to \mathbb{R}^{p \times p}$ creates a diagonal matrix with the elements of an input vector along its diagonal. The operator $\mathsf{diag} : \mathbb{R}^{p \times p} \to \mathbb{R}^p$ extracts the diagonal of a square matrix. For $\boldsymbol{x} \in \mathbb{R}^p$, $\|\boldsymbol{x}\|_\infty = \mathsf{max}_i |x_i|$. $(\boldsymbol{x})^+$ denotes the projection on to the nonnegative orthant, *i.e.*, $(\boldsymbol{x})^+ = \mathsf{max}\,(\boldsymbol{0}, \boldsymbol{x})$.

## 2  Background & Related Works

An undirected, weighted graph is denoted as a triple $\mathcal{G} = (\mathcal{V}, \mathcal{E}, \boldsymbol{W})$, where $\mathcal{V} = \{1, 2, \ldots, p\}$ is the node set, $\mathcal{E} \subseteq \{\{u, v\} : u, v \in \mathcal{V}, u \neq v\}$ is the edge set, that is, a subset of the set of all possible unordered pairs of nodes such that $\{u, v\} \in \mathcal{E}$ iff there exists a link between nodes $u$ and $v$. $\boldsymbol{W} \in \mathbb{R}_+^{p \times p}$ is the symmetric weighted adjacency matrix that satisfies $W_{ii} = 0, W_{ij} > 0$ iff $\{i, j\} \in \mathcal{E}$ and $W_{ij} = 0$, otherwise. The combinatorial, unnormalized graph Laplacian matrix $\boldsymbol{L}$ is defined, as $\boldsymbol{L} \triangleq \boldsymbol{D} - \boldsymbol{W}$, where $\boldsymbol{D} \triangleq \mathsf{Diag}(\boldsymbol{W}\boldsymbol{1})$ is the degree matrix.

A $p$-dimensional, real-valued, Gaussian random variable $\boldsymbol{x}$, with mean vector $\mathbb{E}[\boldsymbol{x}] \triangleq \boldsymbol{\mu}$ and rank-deficient precision matrix $\boldsymbol{L}$, is said to form a Laplacian constrained Gaussian Markov random field (LGMRF) [9, 29–31] of rank $p - k$, $k \geq 1$, with respect to a graph $\mathcal{G}$, when its probability density function is given as

$$p(\boldsymbol{x}) \propto \sqrt{\det{}^*(\boldsymbol{L})} \exp\left\{-\frac{1}{2}(\boldsymbol{x} - \boldsymbol{\mu})^\top \boldsymbol{L}(\boldsymbol{x} - \boldsymbol{\mu})\right\}, \tag{1}$$

where $\det^*(\boldsymbol{L})$ is the pseudo-determinant of $\boldsymbol{L}$, *i.e.*, the product of its positive eigenvalues [32].

Assume we are given $n$ observations of $\boldsymbol{x}$, *i.e.*, $\boldsymbol{X} = [\boldsymbol{x}_1, \boldsymbol{x}_2, \ldots, \boldsymbol{x}_n]^\top$, $\boldsymbol{X} \in \mathbb{R}^{n \times p}$, $\boldsymbol{x}_i \in \mathbb{R}^{p \times 1}$. The goal of graph learning algorithms is to learn a Laplacian matrix, or equivalently an adjacency matrix, given only the data matrix $\boldsymbol{X}$, *i.e.*, often without any knowledge of $\mathcal{E}$.

To that end, the penalized maximum likelihood estimator (MLE) of the Laplacian constrained precision matrix of $\boldsymbol{x}$, on the basis of the observed data $\boldsymbol{X}$, is:

$$\begin{aligned} \underset{\boldsymbol{L} \succeq \boldsymbol{0}}{\mathsf{minimize}} \quad & \mathsf{tr}\,(\boldsymbol{L}\boldsymbol{S}) - \log \det{}^*(\boldsymbol{L}) + h(\boldsymbol{L}), \\ \mathsf{subject\ to} \quad & \boldsymbol{L}\boldsymbol{1} = \boldsymbol{0}, \ L_{ij} = L_{ji} \leq 0, \end{aligned} \tag{2}$$

where $\boldsymbol{S}$ is a similarity matrix, *e.g.*, the sample covariance (or correlation) matrix $\boldsymbol{S} \propto \boldsymbol{X}^\top \boldsymbol{X}$, and $h$ is a regularization function to promote certain properties on $\boldsymbol{L}$ such as sparsity or low-rankness.

Even though Problem (2) is convex, provided we assume a convex choice for $h$, it is not adequate to be solved by disciplined convex programming languages, such as cvxpy [33], particularly due to scalability issues related to the computation of the term $\log \det^*(\boldsymbol{L})$ [6, 7]. Indeed, recently, considerable efforts have been directed towards the design of scalable, iterative algorithms based on block coordinate descent [34], majorization-minimization (MM) [35, 36], and ADMM [37] to solve Problem (2) in an efficient fashion, *e.g.*, [6] and [7].

Estimators based on Gaussian assumptions have been proposed for connected graphs [2, 4–7]. Some of their properties, such as sparsity, are yet being investigated [9, 31]. The authors in [29] and [38] proposed optimization programs for learning the class of $k$-component graphs, as such class is an appealing model for clustering tasks due to the spectral properties of the Laplacian matrix. However, a major shortcoming in their formulations is the lack of constraints on the degrees of the nodes, which allows for trivial solutions, *i.e.*, graphs with isolated nodes.

Elliptical losses along with linear structural constraints that retain the positive-definiteness of the estimated covariance matrix have been proposed in the literature [39]. In this work, however, we address the case of Laplacian constraints, which lead to positive-semidefinite precision matrices, and nonconvex $k$-component structural constraints.

## 3 Proposed Formulations & Algorithms

In this section, we propose optimization formulations and an iterative algorithm to learn a Laplacian matrix from heavy-tailed assumptions. With that goal, we express the Laplacian matrix via its linear operator, *i.e.*, $\boldsymbol{L} = \mathcal{L}\boldsymbol{w}$ [29], where $\boldsymbol{w} \in \mathbb{R}^{p(p-1)/2}$ is the vectorized form of the upper triangular part of the adjacency matrix, also known as the vector of graph weights. In addition, we use the fact that, for connected graphs, it follows that $\det^*(\mathcal{L}\boldsymbol{w}) = \det(\mathcal{L}\boldsymbol{w} + \boldsymbol{J})$, $\boldsymbol{J} \triangleq \frac{1}{p}\boldsymbol{1}\boldsymbol{1}^\top$ [6].

In order to address the inherent heavy-tailed nature of financial market data [40], we consider the Student-$t$ distribution under the improper Markov random field assumption [41] with Laplacian structural constraints, that is, we assume the data generating process to be modeled as multivariate zero-mean Student-$t$ distribution, whose probability density function can be written as

$$p(\boldsymbol{x}) \propto \sqrt{\det^*(\boldsymbol{\Theta})} \left(1 + \frac{\boldsymbol{x}^\top \boldsymbol{\Theta} \boldsymbol{x}}{\nu}\right)^{-\frac{\nu + p}{2}}, \ \nu > 2, \tag{3}$$

where $\boldsymbol{\Theta}$ is a positive-semidefinite inverse scatter matrix modeled as a combinatorial graph Laplacian matrix and $\nu$ is the number of degrees of freedom, which measures the rate of decay of the tails.

This results in a robustified version of the MLE for connected graph learning, *i.e.*,

$$\begin{aligned} \underset{\boldsymbol{w} \geq \boldsymbol{0}, \boldsymbol{\Theta} \succeq \boldsymbol{0}}{\text{minimize}} \quad & \frac{p + \nu}{n} \sum_{i=1}^{n} \log\left(1 + \frac{\boldsymbol{x}_i^\top \mathcal{L}\boldsymbol{w}\boldsymbol{x}_i}{\nu}\right) - \log\det\left(\boldsymbol{\Theta} + \boldsymbol{J}\right), \\ \text{subject to} \quad & \boldsymbol{\Theta} = \mathcal{L}\boldsymbol{w}, \ \mathfrak{d}\boldsymbol{w} = \boldsymbol{d}, \end{aligned} \tag{4}$$

where $\mathfrak{d} : \mathbb{R}^{p(p-1)/2} \to \mathbb{R}^p$ is the degree operator defined as $\mathfrak{d}\boldsymbol{w} \triangleq \text{diag}(\mathcal{L}\boldsymbol{w})$. The constraint $\mathfrak{d}\boldsymbol{w} = \boldsymbol{d}$ enables the learning of additional graph structures such as regular graphs and it is crucial for $k$-component graphs, as discussed in Section 3.2.

From a theoretical perspective, the Student-$t$ model naturally yields sparse graphs. Comparing the objective function in Problem (4) to that of Problem (2), we note that the Student-$t$ contains a $\log(\cdot)$ term in place of a linear term of the graph weights. The usage of a log function to promote sparsity is closely related to the iteratively reweighted $\ell_1$-norm as an approximation for the $\ell_0$-norm problem [42]. Problem (4) is, in general, nonconvex due to the summation of log terms and hence it is challenging to be considered directly. Hence, we design an iterative algorithm based on the ADMM framework.

### 3.1 ADMM Solution

The partial augmented Lagrangian function of Problem (4) is given as

$$\begin{aligned} L_\rho(\boldsymbol{\Theta}, \boldsymbol{w}, \boldsymbol{Y}, \boldsymbol{y}) = & \frac{p + \nu}{n} \sum_{i=1}^{n} \log\left(1 + \frac{\boldsymbol{x}_i^\top \mathcal{L}\boldsymbol{w}\boldsymbol{x}_i}{\nu}\right) - \log\det\left(\boldsymbol{\Theta} + \boldsymbol{J}\right) \\ & + \langle \boldsymbol{y}, \mathfrak{d}\boldsymbol{w} - \boldsymbol{d}\rangle + \frac{\rho}{2}\|\mathfrak{d}\boldsymbol{w} - \boldsymbol{d}\|_2^2 + \langle \boldsymbol{Y}, \boldsymbol{\Theta} - \mathcal{L}\boldsymbol{w}\rangle + \frac{\rho}{2}\|\boldsymbol{\Theta} - \mathcal{L}\boldsymbol{w}\|_{\text{F}}^2, \end{aligned} \tag{5}$$

where $\boldsymbol{Y}$ and $\boldsymbol{y}$ are the dual variables associated with the equality constraints $\boldsymbol{\Theta} = \mathcal{L}\boldsymbol{w}$ and $\mathfrak{d}\boldsymbol{w} = \boldsymbol{d}$, respectively. Note that we deal with the constraints $\boldsymbol{w} \geq \boldsymbol{0}$ and $\boldsymbol{\Theta} \succeq \boldsymbol{0}$ directly, hence there are no dual variables associated with them.

Given $\boldsymbol{w}^l$ and $\boldsymbol{Y}^l$, the subproblem for $\boldsymbol{\Theta}$ can be written as

$$\boldsymbol{\Theta}^{l+1} = \arg\min_{\boldsymbol{\Theta} \succeq \mathbf{0}} \; -\log\det\left(\boldsymbol{\Theta} + \boldsymbol{J}\right) + \langle\boldsymbol{\Theta}, \boldsymbol{Y}^l\rangle + \frac{\rho}{2}\left\|\boldsymbol{\Theta} - \mathcal{L}\boldsymbol{w}^l\right\|_{\mathrm{F}}^2, \tag{6}$$

whose closed-form solution is given by Lemma 1.

**Lemma 1** *The global minimizer of problem* (6) *is* [43, 44]

$$\boldsymbol{\Theta}^{l+1} = \frac{1}{2\rho}\boldsymbol{U}\left(\boldsymbol{\Gamma} + \sqrt{\boldsymbol{\Gamma}^2 + 4\rho\boldsymbol{I}}\right)\boldsymbol{U}^\top - \boldsymbol{J}, \tag{7}$$

*where* $\boldsymbol{U}\boldsymbol{\Gamma}\boldsymbol{U}^\top$ *is the eigenvalue decomposition of* $\rho\left(\mathcal{L}\boldsymbol{w}^l + \boldsymbol{J}\right) - \boldsymbol{Y}^l$.

Given $\boldsymbol{\Theta}^{l+1}$, $\boldsymbol{Y}^l$, and $\boldsymbol{y}^l$, the subproblem for $\boldsymbol{w}$ can be formulated as

$$\operatorname*{minimize}_{\boldsymbol{w}\geq\mathbf{0}} \; \frac{\rho}{2}\boldsymbol{w}^\top\left(\mathfrak{d}^*\mathfrak{d} + \mathcal{L}^*\mathcal{L}\right)\boldsymbol{w} - \left\langle\boldsymbol{w}, \mathcal{L}^*\left(\boldsymbol{Y}^l + \rho\boldsymbol{\Theta}^{l+1}\right) - \mathfrak{d}^*\left(\boldsymbol{y}^l - \rho\boldsymbol{d}\right)\right\rangle$$

$$+ \frac{p+\nu}{n}\sum_{i=1}^n\log\left(1 + \frac{\boldsymbol{x}_i^\top\mathcal{L}\boldsymbol{w}\boldsymbol{x}_i}{\nu}\right), \tag{8}$$

where $\mathfrak{d}^*$ and $\mathcal{L}^*$ are the adjoint operators of the degree and Laplacian operators, respectively.

In general, subproblem (8) is nonconvex due to the concave nature of the logarithm function. Hence, we resort to the MM method [36] to find a stationary point of subproblem (8). We proceed by constructing a global upper-bound of the objective function of (8) at point $\boldsymbol{w}^j \in \mathbb{R}_+^{p(p-1)/2}$ as

$$g(\boldsymbol{w}, \boldsymbol{w}^j) = g(\boldsymbol{w}^j, \boldsymbol{w}^j) + \langle\boldsymbol{w} - \boldsymbol{w}^j, \nabla_{\boldsymbol{w}}f(\boldsymbol{w}^j)\rangle + \frac{\mu}{2}\left\|\boldsymbol{w} - \boldsymbol{w}^j\right\|_2^2, \tag{9}$$

where $f$ is the objective function in the minimization in (8), its gradient is given as $\nabla_{\boldsymbol{w}}f(\boldsymbol{w}^j) = \boldsymbol{a}^j + \boldsymbol{b}^j$, where

$$\boldsymbol{a}^j = \mathcal{L}^*\left(\tilde{\boldsymbol{S}}^j - \boldsymbol{Y}^l - \rho\left(\boldsymbol{\Theta}^{l+1} - \mathcal{L}\boldsymbol{w}^j\right)\right), \tag{10}$$

$$\boldsymbol{b}^j = \mathfrak{d}^*\left(\boldsymbol{y}^l - \rho\left(\boldsymbol{d} - \mathfrak{d}\boldsymbol{w}^j\right)\right), \tag{11}$$

where $\tilde{\boldsymbol{S}}^j \triangleq \frac{1}{n}\sum_{i=1}^n \dfrac{(p+\nu)\boldsymbol{x}_i\boldsymbol{x}_i^\top}{\langle\boldsymbol{w}^j, \mathcal{L}^*(\boldsymbol{x}_i\boldsymbol{x}_i^\top)\rangle + \nu}$ is a weighted sample covariance matrix, and $\mu = \rho\lambda_{\mathsf{max}}\left(\mathfrak{d}^*\mathfrak{d} + \mathcal{L}^*\mathcal{L}\right)$, and the maximum eigenvalue of $\mathfrak{d}^*\mathfrak{d} + \mathcal{L}^*\mathcal{L}$ is given by Lemma 2, whose proof is presented in the Supplementary Material.

**Lemma 2** *The maximum eigenvalue of the matrix* $\mathfrak{d}^*\mathfrak{d} + \mathcal{L}^*\mathcal{L}$ *is given as*

$$\lambda_{\mathsf{max}}\left(\mathfrak{d}^*\mathfrak{d} + \mathcal{L}^*\mathcal{L}\right) = 2(2p-1). \tag{12}$$

The vector of graph weights $\boldsymbol{w}$ can then be updated by minimizing the function $g$ constructed in (9), which is tantamount to solving the following nonnegative, quadratic-constrained, strictly convex problem:

$$\boldsymbol{w}^{j+1} = \arg\min_{\boldsymbol{w}\geq\mathbf{0}} \rho(2p-1)\left\|\boldsymbol{w} - \boldsymbol{w}^j\right\|_2^2 + \langle\boldsymbol{w}, \boldsymbol{a}^j + \boldsymbol{b}^j\rangle, \tag{13}$$

whose unique solution can be readily obtained via its KKT optimality conditions and is given as

$$\boldsymbol{w}^{j+1} = \left(\boldsymbol{w}^j - \frac{\boldsymbol{a}^j + \boldsymbol{b}^j}{2\rho(2p-1)}\right)^+, \tag{14}$$

that is, a projected gradient descent step with learning rate $(2\rho(2p-1))^{-1}$. Thus, we iterate (14) in order to obtain a stationary point, $\boldsymbol{w}^{l+1}$, of Problem (8). In practice, we observe that a few iterations are sufficient to retrieve $\boldsymbol{w}^{l+1}$.

The dual variables $\boldsymbol{Y}$ and $\boldsymbol{y}$ are updated via gradient ascent steps. Algorithm 1 summarizes the implementation to find a stationary point of Problem (4). We present Theorem 3, proved in the Supplementary Material, which establishes the convergence of Algorithm 1.

**Theorem 3** *Algorithm 1 converges subsequently for any sufficiently large $\rho$, that is, the sequence* $\left\{\left(\boldsymbol{\Theta}^l, \boldsymbol{w}^l, \boldsymbol{Y}^l, \boldsymbol{y}^l\right)\right\}$ *generated by Algorithm 1 has at least one limit point, and each limit point is a stationary point of Problem (4).*

---

**Algorithm 1:** Student-$t$ Graph Learning

---

**Data:** Data matrix $\boldsymbol{X} \in \mathbb{R}^{n \times p}$, initial estimate of the graph weights $\boldsymbol{w}^0$, desired degree vector $\boldsymbol{d}$, penalty parameter $\rho > 0$, degrees of freedom $\nu > 2$, convergence tolerance $\epsilon > 0$

**Result:** Graph Laplacian estimation: $\mathcal{L}\boldsymbol{w}^\star$

**1** initialize $\boldsymbol{Y} = \boldsymbol{0}, \boldsymbol{y} = \boldsymbol{0}$

**2** $l \leftarrow 0$

**3** **while** $\left\| \boldsymbol{r}^l \right\|_\infty > \epsilon$ *or* $\left\| \boldsymbol{s}^l \right\|_\infty > \epsilon$ **do**

**4**     $\triangleright$ update $\boldsymbol{\Theta}^{l+1}$ via (7)

**5**     $\triangleright$ update $\boldsymbol{w}^{l+1}$ by iterating (14)

**6**     $\triangleright$ update $\boldsymbol{Y}^{l+1} = \boldsymbol{Y}^l + \rho \left( \boldsymbol{\Theta}^{l+1} - \mathcal{L}\boldsymbol{w}^{l+1} \right)$

**7**     $\triangleright$ update $\boldsymbol{y}^{l+1} = \boldsymbol{y}^l + \rho \left( \mathfrak{d}\boldsymbol{w}^{l+1} - \boldsymbol{d} \right)$

**8**     $\triangleright$ compute residual $\boldsymbol{r}^{l+1} = \boldsymbol{\Theta}^{l+1} - \mathcal{L}\boldsymbol{w}^{l+1}$

**9**     $\triangleright$ compute residual $\boldsymbol{s}^{l+1} = \mathfrak{d}\boldsymbol{w}^{l+1} - \boldsymbol{d}$

**10**     $l \leftarrow l + 1$

**11** **end**

---

## 3.2   An extension to $k$-component graphs

The graph learning formulation proposed in (4) is applicable to learn connected graphs. Learning graphs with $k$ components, $k$ assumed to be known, poses a considerably higher challenge, as the dimension of the nullspace of the Laplacian matrix $\mathcal{L}\boldsymbol{w}$ is equal to the number of components of the graph [45]. One way to achieve this requirement is by imposing the constraint $\mathsf{rank}\,(\mathcal{L}\boldsymbol{w}) = p - k$, which is nonconvex and nondifferentiable, in the maximum likelihood problem generated by (3). We instead relax this rank constraint by noting that via Fan's theorem [46], we have

$$\sum_{i=1}^{k} \lambda_i\,(\mathcal{L}\boldsymbol{w}) = \underset{\boldsymbol{V} \in \mathbb{R}^{p \times k}, \boldsymbol{V}^\top \boldsymbol{V} = \boldsymbol{I}}{\text{minimize}} \mathsf{tr}\left(\boldsymbol{V}^\top \mathcal{L}\boldsymbol{w}\boldsymbol{V}\right). \tag{15}$$

Thus, by using the right hand side of (15) as a regularization term, we are able formulate the following optimization problem to learn a Student-$t$ $k$-component graph:

$$\begin{aligned}
&\underset{\boldsymbol{w} \geq \boldsymbol{0}, \boldsymbol{\Theta} \succeq \boldsymbol{0}, \boldsymbol{V}}{\text{minimize}} \quad \frac{p + \nu}{n} \sum_{i=1}^{n} \log\left(1 + \frac{\boldsymbol{x}_i^\top \mathcal{L}\boldsymbol{w}\boldsymbol{x}_i}{\nu}\right) - \log \det{}^*(\boldsymbol{\Theta}) + \eta\mathsf{tr}(\mathcal{L}\boldsymbol{w}\boldsymbol{V}\boldsymbol{V}^\top), \\
&\text{subject to} \quad \boldsymbol{\Theta} = \mathcal{L}\boldsymbol{w}, \; \mathsf{rank}(\boldsymbol{\Theta}) = p - k, \mathfrak{d}\boldsymbol{w} = \boldsymbol{d}, \; \boldsymbol{V}^\top \boldsymbol{V} = \boldsymbol{I}, \; \boldsymbol{V} \in \mathbb{R}^{p \times k}.
\end{aligned} \tag{16}$$

The partial augmented Lagrangian function of Problem (16) can be expressed as

$$\begin{aligned}
L_\rho(\boldsymbol{\Theta}, \boldsymbol{w}, \boldsymbol{V}, \boldsymbol{Y}, \boldsymbol{y}) = {}& \frac{p + \nu}{n} \sum_{i=1}^{n} \log\left(1 + \frac{\boldsymbol{x}_i^\top \mathcal{L}\boldsymbol{w}\boldsymbol{x}_i}{\nu}\right) - \log \det{}^*(\boldsymbol{\Theta}) + \eta\mathsf{tr}\left(\mathcal{L}\boldsymbol{w}\boldsymbol{V}\boldsymbol{V}^\top\right) \\
&+ \langle \boldsymbol{y}, \mathfrak{d}\boldsymbol{w} - \boldsymbol{d} \rangle + \frac{\rho}{2} \|\mathfrak{d}\boldsymbol{w} - \boldsymbol{d}\|_2^2 + \langle \boldsymbol{Y}, \boldsymbol{\Theta} - \mathcal{L}\boldsymbol{w} \rangle + \frac{\rho}{2} \|\boldsymbol{\Theta} - \mathcal{L}\boldsymbol{w}\|_{\mathrm{F}}^2. \tag{17}
\end{aligned}$$

Given $\boldsymbol{w}^l$ and $\boldsymbol{Y}^l$, the subproblem for $\boldsymbol{\Theta}$ can be written as

$$\boldsymbol{\Theta}^{l+1} = \underset{\mathsf{rank}(\boldsymbol{\Theta}) = p - k}{\arg\min} \; -\log \det{}^*(\boldsymbol{\Theta}) + \langle \boldsymbol{\Theta}, \boldsymbol{Y}^l \rangle + \frac{\rho}{2} \|\boldsymbol{\Theta} - \mathcal{L}\boldsymbol{w}^l\|_{\mathrm{F}}^2, \tag{18}$$

which is nearly the same as (6). Its solution is obtained as

$$\boldsymbol{\Theta}^{l+1} = \frac{1}{2\rho} \boldsymbol{U}\left(\boldsymbol{\Gamma} + \sqrt{\boldsymbol{\Gamma}^2 + 4\rho\boldsymbol{I}}\right) \boldsymbol{U}^\top, \tag{19}$$

except that now $\boldsymbol{U}\boldsymbol{\Gamma}\boldsymbol{U}^\top$ is the eigenvalue decomposition of $\rho\mathcal{L}\boldsymbol{w}^l - \boldsymbol{Y}^l$, with $\boldsymbol{\Gamma}$ having the largest $p - k$ eigenvalues along its diagonal and $\boldsymbol{U} \in \mathbb{R}^{p \times (p-k)}$ contains the corresponding eigenvectors.

The subproblem to obtain $\boldsymbol{w}^{l+1}$ is virtually the same as in (8) except for the additional linear term $\eta\mathsf{tr}(\mathcal{L}\boldsymbol{w}\boldsymbol{V}^l\boldsymbol{V}^{l\top})$. Hence, its update is also a projected gradient descent step, alike (14) where

$$\boldsymbol{a}^j \triangleq \mathcal{L}^*\left(\tilde{\boldsymbol{S}}^j + \eta\boldsymbol{V}^l\boldsymbol{V}^{l\top} - \boldsymbol{Y}^l - \rho\left(\boldsymbol{\Theta}^{l+1} - \mathcal{L}\boldsymbol{w}^j\right)\right). \tag{20}$$

Given $\boldsymbol{w}^{l+1}$, we have the following subproblem for $\boldsymbol{V}$:

$$\underset{\boldsymbol{V}\in\mathbb{R}^{p\times k},\boldsymbol{V}^{\top}\boldsymbol{V}=\boldsymbol{I}}{\text{minimize}}\quad \mathsf{tr}\left(\boldsymbol{V}^{\top}\mathcal{L}\boldsymbol{w}^{l+1}\boldsymbol{V}\right), \tag{21}$$

whose closed-form solution is given by the $k$ eigenvectors associated with the $k$ smallest eigenvalues of $\mathcal{L}\boldsymbol{w}^{l+1}$ [47, 48]. Algorithm 2 summarizes the implementation to find a stationary point of Problem (16), and its convergence is established through Theorem 4, whose proof is presented in the Supplementary Material.

**Theorem 4** *Algorithm 2 converges subsequently for any sufficiently large $\rho$, that is, the sequence $\left\{\left(\boldsymbol{\Theta}^{l},\boldsymbol{w}^{l},\boldsymbol{V}^{l},\boldsymbol{Y}^{l},\boldsymbol{y}^{l}\right)\right\}$ generated by Algorithm 2 has at least one limit point, and each limit point is a stationary point of Problem (16).*

---

**Algorithm 2:** $k$-component Student-$t$ graph learning

---

**Data:** Data matrix $\boldsymbol{X}\in\mathbb{R}^{n\times p}$, initial estimate of the graph weights $\boldsymbol{w}^{0}$, number of graph components $k$, desired degree vector $\boldsymbol{d}$, degrees of freedom $\nu$, rank hyperparameter $\eta>0$, penalty parameter $\rho>0$, tolerance $\epsilon>0$

**Result:** Laplacian estimation: $\mathcal{L}\boldsymbol{w}^{\star}$

1 initialize $\boldsymbol{Y}=\boldsymbol{0},\boldsymbol{y}=\boldsymbol{0}$
2 $l\leftarrow 0$
3 **while** $\left\|\boldsymbol{r}^{l}\right\|_{\infty}>\epsilon$ *or* $\left\|\boldsymbol{s}^{l}\right\|_{\infty}>\epsilon$ **do**
4     ▷ update $\boldsymbol{\Theta}^{l+1}$ via (19)
5     ▷ update $\boldsymbol{w}^{l+1}$ as in (14) with $\boldsymbol{a}^{j}$ given in (20)
6     ▷ update $\boldsymbol{V}^{l+1}$ as in (21)
7     ▷ update $\boldsymbol{Y}^{l+1}=\boldsymbol{Y}^{l}+\rho\left(\boldsymbol{\Theta}^{l+1}-\mathcal{L}\boldsymbol{w}^{l+1}\right)$
8     ▷ update $\boldsymbol{y}^{l+1}=\boldsymbol{y}^{l}+\rho\left(\mathfrak{d}\boldsymbol{w}^{l+1}-\boldsymbol{d}\right)$
9     ▷ compute residual $\boldsymbol{r}^{l+1}=\boldsymbol{\Theta}^{l+1}-\mathcal{L}\boldsymbol{w}^{l+1}$
10     ▷ compute residual $\boldsymbol{s}^{l+1}=\mathfrak{d}\boldsymbol{w}^{l+1}-\boldsymbol{d}$
11     $l\leftarrow l+1$
12 **end**

---

## 4 Experiments

To evaluate the performance of the proposed graph learning algorithms, we perform experiments using historical daily price time series data, available in Yahoo! Finance™, from financial instruments in three scenarios: (i) stocks belonging to the S&P500 index, (ii) foreign exchange markets, and (iii) cryptocurrencies. We start by constructing the log-returns data matrix, *i.e.*, a matrix $\boldsymbol{X}\in\mathbb{R}^{n\times p}$, where $n$ is the number of log-return observations and $p$ is the number of instruments, as

$$X_{i,j}=\log P_{i,j}-\log P_{i-1,j}, \tag{22}$$

where $P_{i,j}$ is the closing price of the $j$-th instrument at the $i$-th day.

*Benchmarks*: We compare our proposed algorithms with state-of-the-art, Gaussian distribution-based methods for connected graphs, namely GLE [7] and NGL [9], which use $\ell_1$-norm and minimax concave penalty regularizations, respectively; and CLR [38] and SGL [29] that consider $k$-component graphs. For a fair comparison among algorithms, we set the degree vector $\boldsymbol{d}$ equal to $\boldsymbol{1}$ for the proposed algorithms, *i.e.*, we do not consider any prior information on the degree of nodes. In our ADMM algorithms, we set the penalty parameter to $\rho=1$ and the hyperparameter $\eta$ in (16) is adaptively increased until the rank constraint is satisfied. For GLE and NGL, we use grid search on the sparsity hyperparameter such that the resulting graph yields the highest modularity value. The graph weights in Algorithm 1 and 2 are initialized using the same procedure as in [8].

Our goal with the experiments that follow is to verify whether the heavy-tail assumption provides an improved version of the learned graph, which is evaluated based on the modularity[1] of the estimated

---

[1]Modularity measures the strength of separability of a graph into groups [49].

graph and the graph visualization. In addition, for the task of clustering stocks, we analyze whether the learned graphs agree with industry standards of sector classification set by the Global Industry Classification Standard (GICS) [50, 51]. [2]

## 4.1 Communities in S&P500 Stocks

In this experiment, we consider S&P500 stocks belonging to three sectors, namely, Communication Services (red), Utilities (blue), and Real Estate (green), totalling $p = 82$ stocks, during the time horizon from Jan. 3rd 2014 to Dec. 29th 2017, resulting in $n = 1006$ observations. In order to obtain descriptive insights on this dataset, we measure its degree of heavy-tailedness and annualized volatility[3]. The former is obtained by fitting the degrees of freedom of a Student-$t$ distribution to the matrix of log-returns, whereby we obtain $\nu \approx 5.5$ and $\sigma \approx 21\%$. This scenario can be considered as having a moderate amount of heavy-tailedness.

Figure 1 depicts the learned connected graphs on the aforementioned time periods. It can be readily noticed that the graph learned with the Student-$t$ distribution (Figure 1c) is sparser than those learned with the Gaussian assumption (Figure 1a and 1b), which results from the fact that the Gaussian distribution is more sensitive to outliers. Moreover, the Student-$t$ graph presents a higher degree of interpretability as measured by its modularity value. In addition, a larger number of inter-sector connections, as indicated by gray-colored edges, which are often spurious from a practical perspective, are present in the graphs learned by NGL and GLE. Sparsity regularization provides a means to remove edges between nodes in the presence of data with outliers and possibly increasing the modularity of the resulting graph. However, they bring the additional task of tunning hyperparameters, which is often repetitive and impractical for real-time applications. A cleaner graph, without the need for postprocessing or additional regularization, is obtained directly by using the Student-$t$ assumption.

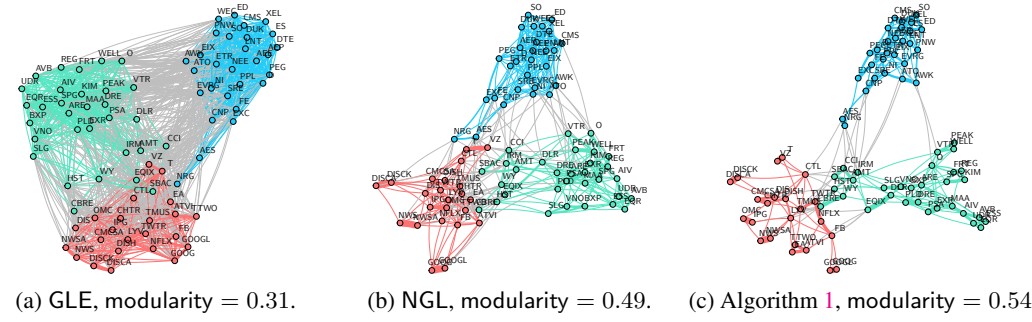

(a) GLE, modularity = 0.31.    (b) NGL, modularity = 0.49.    (c) Algorithm 1, modularity = 0.54.

Figure 1: Learned graphs of S&P500 stocks.

Figure 2 illustrates the learned 3-component graphs during the time from Jan. 3rd 2014 until Dec. 29th 2017. We can notice that SGL (Figure 2a) and CLR (Figure 2b) are unable to separate the stocks in a way that agrees with their sector information as given by GICS. In addition, the high number of spurious connections (gray-colored edges) are uncharacteristic of the actual expected behavior in stock markets. Figure 2c displays the graph learned by the proposed Algorithm 2, where it presents not only a higher modularity value, but also a sparser, more plausible representation of an actual network of stocks with three sectors.

### 4.1.1 Impact of COVID-19 in the US Stock Market

We collect price data of $p = 85$ stocks belonging to three sectors, namely, Communication Services (red), Utilities (blue), and Real Estate (green) from Apr. 22nd 2019 to Dec. 30th 2020, resulting in $n = 429$ observations. The degrees of freedom and annualized volatility during this period, which includes the financial crisis caused by the COVID-19 pandemic, were $\nu \approx 2.89$ and $\sigma \approx 41\%$, indicating a strongly heavy-tailed scenario.

---

[2]More often than not, stocks have impacts on multiple sectors, *e.g.*, the evident case of technology companies, whose influence affect the prices of stocks not only in their own sector, but spans across multiple sectors. Therefore, GICS or other sector classification systems cannot be considered as absolute ground-truth labels.

[3]The annualized volatility is computed as $\sigma = \frac{\sqrt{252}}{p} \sum_{i=1}^{p} \sigma_i$, where $\sigma_i$ is the daily sample standard deviation of the $i$-th stock.

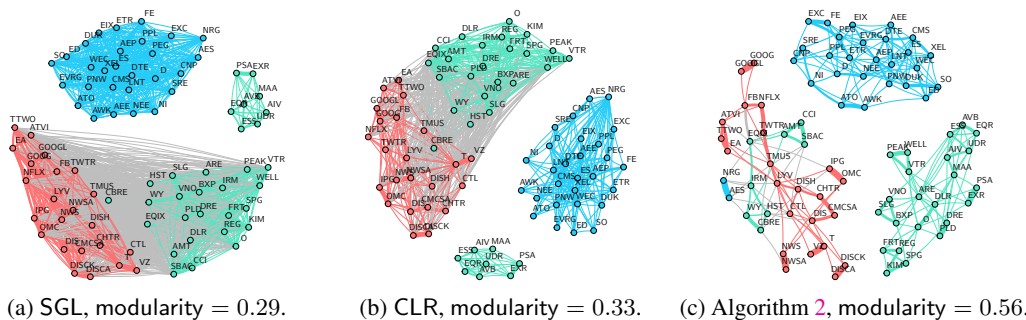

(a) SGL, modularity = 0.29.     (b) CLR, modularity = 0.33.     (c) Algorithm 2, modularity = 0.56.

Figure 2: Learned 3-component graphs of S&P500 stocks.

In Figure 3, we observe that the modularity value of the Gaussian-based graphs are severely reduced as compared to the results presented in Figure 1, while the opposite occurs with the Student-$t$ graph. That may be explained by the increase in variability and outliers in the data during the financial crisis caused by the COVID-19 pandemic, which is better modelled by a Student-$t$ distribution. In addition, the learned graph in Figure 3c clearly displays expected behaviors such as the strong correlation between pairs of companies including {Zoom Video Communications Inc., Netflix Inc.}, and {Twitter Inc., Facebook Inc.}, which are not as evident in Figures 3a and 3b.

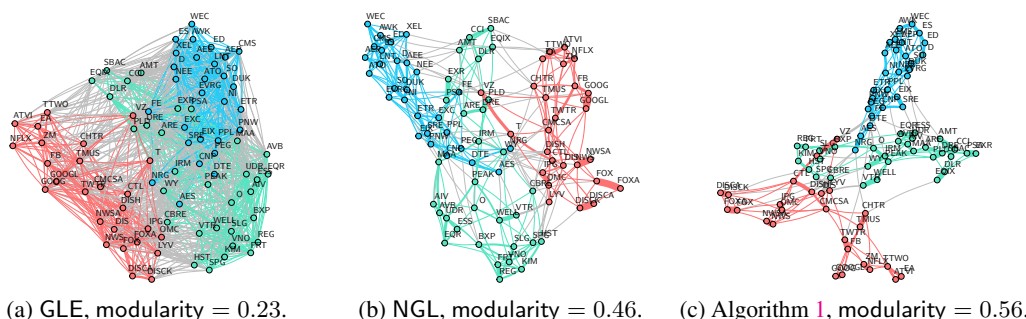

(a) GLE, modularity = 0.23.     (b) NGL, modularity = 0.46.     (c) Algorithm 1, modularity = 0.56.

Figure 3: Learned graphs of S&P500 stocks during the COVID-19 pandemic.

## 4.2 Communities in Foreign Exchange Markets

We query foreign exchange data from the 34 most traded currencies between the period from Jan. 2nd 2019 to Dec. 31st 2020, totalling $n = 522$ observations. The data matrix is composed by the log-returns of the currencies prices with respect to the United States Dollar. Similar to the previous experiment, we compute the degrees of freedom of a Student-$t$ distribution fitted to the log-returns data matrix, whereby we obtain $\nu \approx 4.6$, which represents a scenario with considerable amount of heavy-tailedness. Unlike in the experiment involving S&P500 stocks, there are no classification standard for currencies, hence we rely on a community detection algorithm [52] in order to create classes within the learned graph. In particular, the algorithm in [52] takes as input the learned Laplacian matrix of the graph and outputs a membership assignment that maximizes the modularity of the graph.

Figure 4 displays the learned graphs. As it can be observed, the Student-$t$ graph (Figure 4c) is sparser, more interpretable, and has a higher modularity value than that of the Gaussian-based graphs (Figure 4a and 4b). In addition, the expected correlation between currencies of locations geographically close to each other, *e.g.*, {Hong Kong SAR, China}, {Taiwan, South Korea}, and {Poland, Czech Republic} are significantly more evident for the Student-$t$ graph.

Figure 5 depicts the learned 9-component graphs of currencies during the time window from Jan. 2nd 2019 to Dec. 31st 2020. It can be observed that the graph learned by the proposed Algorithm 2 (Figure 5c) presents a finer structure and a higher modularity value than those learned by SGL (Figure 5a) and CLR (Figure 5b). In addition, the learned graph in Figure 5c presents more reasonable clusters such as {New Zealand, Australia} and {Poland, Czech Republic, Hungary}, which are not

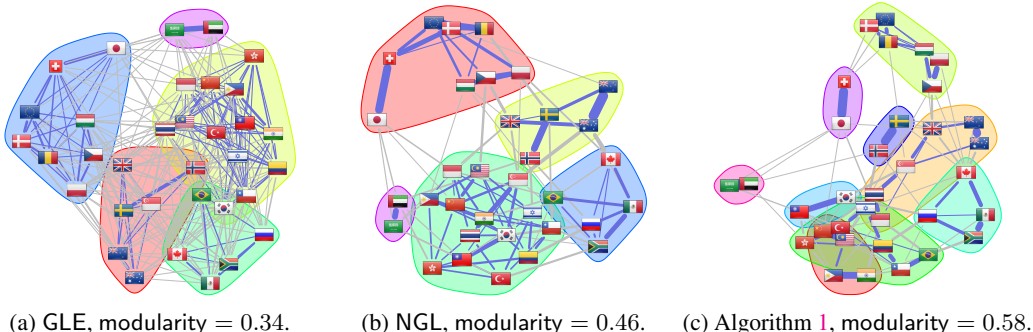

(a) GLE, modularity = 0.34.    (b) NGL, modularity = 0.46.    (c) Algorithm 1, modularity = 0.58.

Figure 4: Learned connected graphs of currencies.

separated in the Gaussian-based graphs. More critically, we can observe that SGL and CLR allow the existence of isolated nodes in the learned graphs. In our proposed algorithm, we avoid such solutions by imposing linear constraints on the degree of the graph.

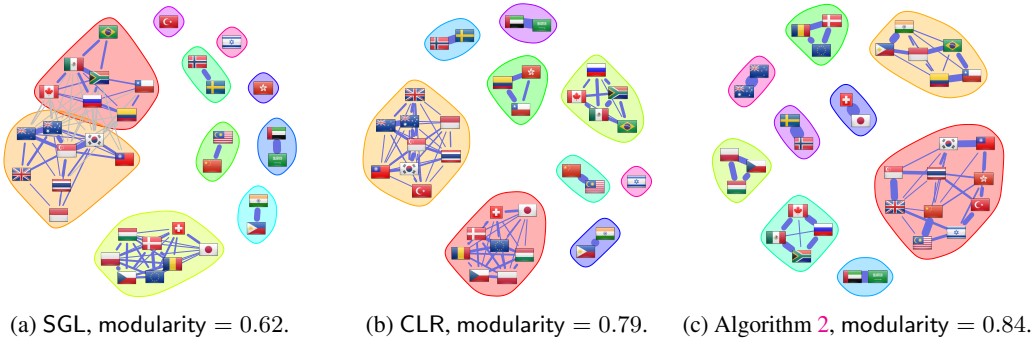

(a) SGL, modularity = 0.62.    (b) CLR, modularity = 0.79.    (c) Algorithm 2, modularity = 0.84.

Figure 5: Learned 9-component graphs of currencies.

## 4.3 Communities in Cryptocurrencies

We query daily prices of the $p = 41$ most traded cryptocurrencies during the period starting from Aug. 1st 2017 to Dec. 1st 2020, which amounts to $n = 1218$ observations. The degrees of freedom during this time frame was measured as $\nu \approx 3$, which is tantamount to a strong heavy-tail scenario.

Figure 6 shows the learned graphs by GLE, NGL, and our proposed Algorithm 1. While the graphs in Figures 6a and 6b present a small modularity value and contain a large number of edges, which impairs interpretability, the resulting proposed graph in Figure 6c reveals a refined representation of the interactions between pairs of cryptocurrencies, which is possibly more aligned with the actual market scenario. As an example, the link between Bitcoin (BTC) and Litecoin (LTC), a Bitcoin spinoff established in 2011, is substantially more evident in Figure 6c.

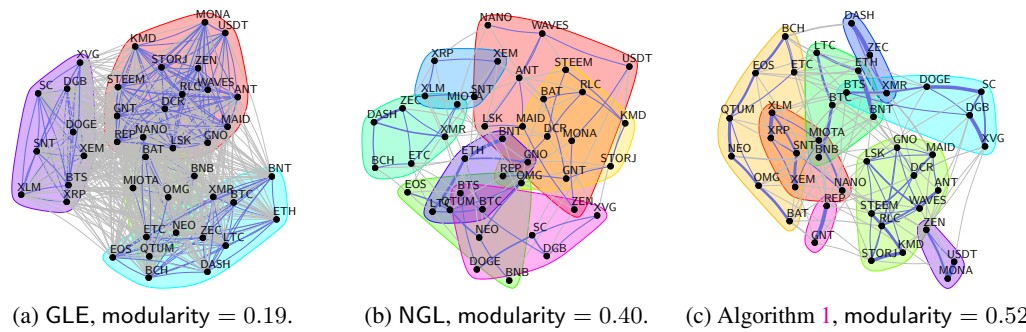

(a) GLE, modularity = 0.19.    (b) NGL, modularity = 0.40.    (c) Algorithm 1, modularity = 0.52.

Figure 6: Learned connected graphs of cryptocurrencies.

Figure 7 shows the learned 7-component graphs of cryptocurrencies during the aforementioned time window. As in the previous experiments with foreign exchange data, SGL shows isolated nodes in the learned graph (Figure 7a) and CLR contains a large number of spurious connections in the main cluster (Figure 7b), whereas the graph learned via Algorithm 2 (Figure 7c) has the largest modularity value. Interestingly, while all three methods agree to cluster {Dogecoin (DOGE), Verge (XVG), Siacoin (SC), DigiByteCoin (DGB)}, which may be related to their similar initial release dates, only our proposed algorithm clusters together the coins that mainly focus on privacy and anonymity features, *i.e.*, {Monero (XMR), Zcash (ZEC), DASH}.

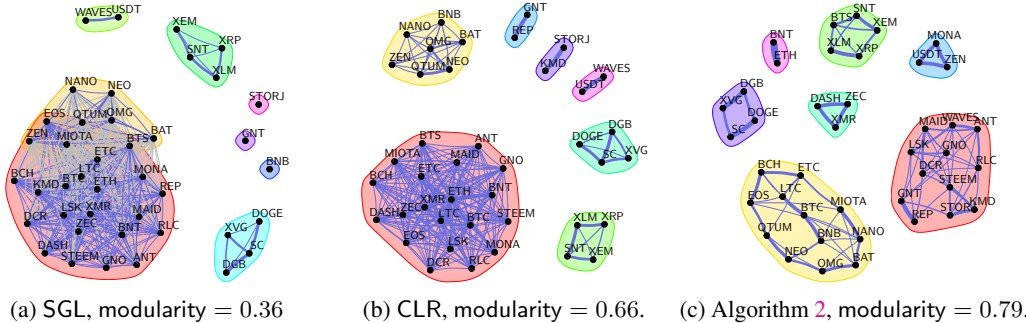

| (a) SGL, modularity $= 0.36$ | (b) CLR, modularity $= 0.66$. | (c) Algorithm 2, modularity $= 0.79$. |

Figure 7: Learned 7-component graphs of cryptocurrencies.

## 4.4 Effect of Algorithm Initialization

Initialization is a critical phase in any iterative algorithm, especially when dealing with nonconvex optimization problems involving nonconvex constraints as it is the case of Algorithm 2. To verify the stability of Algorithm 2, we initialize it as a fully connected graph with equal weights, *i.e.*, $w^0 \propto 1$. Figure 8 illustrates that there is no significant difference from the estimated graphs with uniform initial point and the ones reported in the previous section. This result provides evidence that Algorithm 2 is robust against inaccurate initializations.

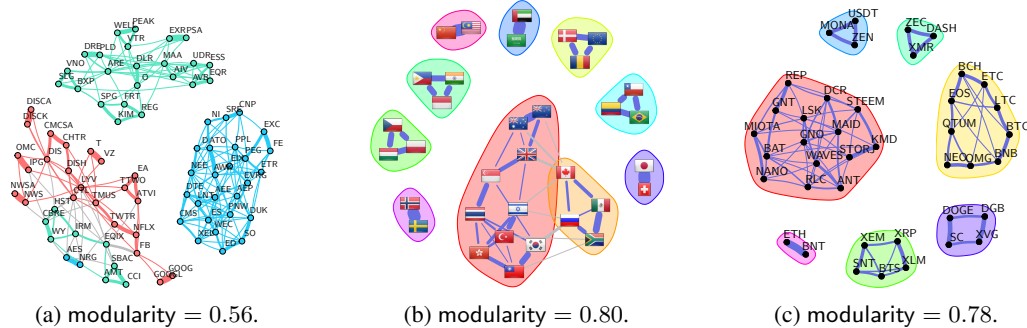

| (a) modularity $= 0.56$. | (b) modularity $= 0.80$. | (c) modularity $= 0.78$. |

Figure 8: Learned graphs by Algorithm 2 with an uniform initial point.

## 5 Conclusions

Heavy-tails are prevalent in time-series of financial markets. Yet, they have been little explored in the context of Laplacian graphical models. In this paper, we have proposed optimization programs to learn graphical models with Laplacian constraints assuming that the data generating process is Student-$t$ distributed. The formulations follow a maximum likelihood approach of a Markov random field, for which we designed ADMM algorithms that converge to a stationary point of the resulting nonconvex problems. The proposed algorithms showed significant gains, measured via the modularity values of the estimated graphs, when compared to state-of-the-art counterparts in real-world scenarios that involved data from the US stock market, foreign exchange markets, and cryptocurrencies.

## Acknowledgments

We would like to thank the anonymous reviewers for their helpful comments. This work was supported by the Hong Kong GRF 16207820 research grant.

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
