# Supplementary Material

In this section, we present definitions that were not explicitly stated in the main manuscript, alongside additional validation experiments that showcase the robustness of our algorithms, and theoretical proofs of Lemma 2 and Theorems 3 and 4.

## A    Definitions

**Definition 5** *(**Laplacian operator**) The Laplacian operator [8] $\mathcal{L} : \mathbb{R}_+^{p(p-1)/2} \to \mathbb{R}^{p \times p}$, which takes a nonnegative vector $\boldsymbol{w}$ and outputs a Laplacian matrix $\mathcal{L}\boldsymbol{w}$, is defined as*

$$[\mathcal{L}\boldsymbol{w}]_{ij} = \begin{cases} -w_{i+s(j)}, & \text{if } i > j, \\ [\mathcal{L}\boldsymbol{w}]_{ji}, & \text{if } i < j, \\ -\sum_{i \neq j}[\mathcal{L}\boldsymbol{w}]_{ij}, & \text{if } i = j, \end{cases} \tag{23}$$

*where $s(j) = \frac{j-1}{2}(2p-j) - j$.*

**Definition 6** *(**Degree operator**) The degree operator $\mathfrak{d} : \mathbb{R}^{p(p-1)/2} \to \mathbb{R}^p$, which takes a nonnegative vector $\boldsymbol{w}$ and outputs the diagonal of a Degree matrix, is defined as*

$$\mathfrak{d}\boldsymbol{w} = \text{diag}(\mathcal{L}\boldsymbol{w}). \tag{24}$$

**Definition 7** *(**Adjoint of Laplacian operator**) The adjoint of Laplacian operator [8] $\mathcal{L}^* : \mathbb{R}^{p \times p} \to \mathbb{R}^{p(p-1)/2}$ is defined as*

$$(\mathcal{L}^*\boldsymbol{P})_{s(i,j)} = \boldsymbol{P}_{i,i} - \boldsymbol{P}_{i,j} - \boldsymbol{P}_{j,i} + \boldsymbol{P}_{j,j} \tag{25}$$

*where $s(i,j) = i - j + \frac{i-1}{2}(2p-j), i > j$.*

**Definition 8** *(**Adjoint of degree operator**) The adjoint of degree operator $\mathfrak{d}^* : \mathbb{R}^p \to \mathbb{R}^{p(p-1)/2}$ is given as*

$$(\mathfrak{d}^*\boldsymbol{y})_{s(i,j)} = \boldsymbol{y}_i + \boldsymbol{y}_j, \tag{26}$$

*where $s(i,j) = i - j + \frac{i-1}{2}(2p-j), i > j$.*

**Definition 9** *(**Modularity**) The modularity of a graph $\mathcal{G}$ [49] is defined as $Q : \mathcal{G} \to \mathbb{R}$:*

$$Q(\mathcal{G}) \triangleq \frac{1}{2|\mathcal{E}|} \sum_{i,j \in \mathcal{V}} \left( \boldsymbol{W}_{ij} - \frac{d_i d_j}{2|\mathcal{E}|} \right) \mathbb{1}(t_i = t_j), \tag{27}$$

*where $d_i$ is the weighted degree of the $i$-th node, i.e. $d_i \triangleq [\mathfrak{d}(\boldsymbol{w})]_i$, $t_i$ is the type (or label) of the $i$-th node, and $\mathbb{1}(\cdot)$ is the indicator function.*

## B    Additional Experiments

### B.1    Empirical Convergence

In this section, we illustrate the empirical convergence performance of the proposed algorithms. All the experiments were carried out in a MacBook Pro 13in. 2019 with Intel Core i7 2.8GHz, 16GB of RAM. In Figure 9, we observe that the Lagrangian function quickly approaches a stationary value after a transient phase typical of ADMM-like algorithms.

## C    Proofs

### C.1    Proof of Lemma 2

**Proof** We define an index set $\Omega_t$:

$$\Omega_t \triangleq \left\{ l : [\mathcal{L}\boldsymbol{x}]_{tt} = \sum_{l \in \Omega_t} x_l \right\}, \quad t \in [1, 2, \dots, p]. \tag{28}$$

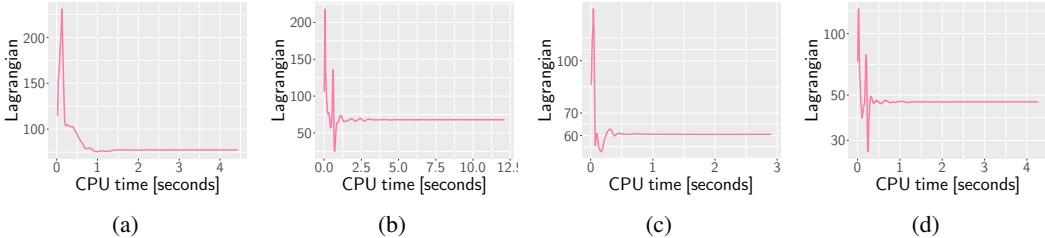

Figure 9: Empirical convergence for Algorithm 1 (panels (a) and (c)), *i.e.* connected graphs, and Algorithm 2 (panels (b) and (d)), *i.e.*, $k$-component graphs, with cryptocurrencies data (panels (a) and (b)) and foreign exchange data (panels (c) and (d)).

Then we have

$$\lambda_{\max}\left(\mathcal{L}^*\mathcal{L}\right) = \sup_{\|\boldsymbol{x}\|=1} \boldsymbol{x}^\top \mathcal{L}^*\mathcal{L}\boldsymbol{x} = \sup_{\|\boldsymbol{x}\|=1} \|\mathcal{L}\boldsymbol{x}\|_{\mathrm{F}}^2 = \sup_{\|\boldsymbol{x}\|=1} 2\sum_{k=1}^{p(p-1)/2} x_k^2 + \sum_{i=1}^{p}([\mathcal{L}\boldsymbol{x}]_{ii})^2$$

$$= \sup_{\|\boldsymbol{x}\|=1} 4\sum_{k=1}^{p(p-1)/2} x_k^2 + \sum_{t=1}^{p}\sum_{i,j\in\Omega_t,\ i\neq j} x_i x_j \le 4 + \sup_{\|\boldsymbol{x}\|=1} \frac{1}{2}\sum_{t=1}^{p}\sum_{i,j\in\Omega_t,\ i\neq j} x_i^2 + x_j^2$$

$$= (4 + 2(|\Omega_t| - 1)) \sup_{\|\boldsymbol{x}\|=1} \sum_{k=1}^{p(p-1)/2} x_k^2 = 2p,$$

with equality if and only if $x_1 = \cdots = x_{p(p-1)/2} = \sqrt{\frac{2}{p(p-1)/2}}$ or $x_1 = \cdots = x_{p(p-1)/2} = -\sqrt{\frac{2}{p(p-1)/2}}$. The last equality follows the fact that $|\Omega_t| = p - 1$.

Similarly, we can obtain

$$\lambda_{\max}\left(\mathfrak{d}^*\mathfrak{d}\right) = \sup_{\|\boldsymbol{x}\|=1} \boldsymbol{x}^\top \mathfrak{d}^*\mathfrak{d}\boldsymbol{x} = \sup_{\|\boldsymbol{x}\|=1} \|\mathfrak{d}\boldsymbol{x}\|^2 = \sup_{\|\boldsymbol{x}\|=1} 2\sum_{k=1}^{p(p-1)/2} x_k^2 + \sum_{t=1}^{p}\sum_{i,j\in\Omega_t,\ i\neq j} x_i x_j$$

$$\le 2 + \sup_{\|\boldsymbol{x}\|=1} \frac{1}{2}\sum_{t=1}^{p}\sum_{i,j\in\Omega_t,\ i\neq j} x_i^2 + x_j^2 = (2 + 2(|\Omega_t| - 1)) \sup_{\|\boldsymbol{x}\|=1} \sum_{k=1}^{p(p-1)/2} x_k^2 = 2p - 2,$$

with equality if and only if $x_1 = \cdots = x_{p(p-1)/2} = \sqrt{\frac{2}{p(p-1)/2}}$ or $x_1 = \cdots = x_{p(p-1)/2} = -\sqrt{\frac{2}{p(p-1)/2}}$.

Finally, we have

$$\begin{aligned}\lambda_{\max}\left(\mathcal{L}^*\mathcal{L} + \mathfrak{d}^*\mathfrak{d}\right) &= \sup_{\|\boldsymbol{x}\|=1} \boldsymbol{x}^\top \left(\mathcal{L}^*\mathcal{L} + \mathfrak{d}^*\mathfrak{d}\right)\boldsymbol{x} \\ &\le \sup_{\|\boldsymbol{x}\|=1} \boldsymbol{x}^\top \left(\mathcal{L}^*\mathcal{L}\right)\boldsymbol{x} + \sup_{\|\boldsymbol{y}\|=1} \boldsymbol{y}^\top \left(\mathfrak{d}^*\mathfrak{d}\right)\boldsymbol{y} \qquad (29) \\ &= 4p - 2.\end{aligned}$$

Note that the equality in (29) can be achieved because the eigenvectors of $\mathcal{L}^*\mathcal{L}$ and $\mathfrak{d}^*\mathfrak{d}$ associated with the maximum eigenvalue are the same. Therefore, we conclude that $\lambda_{\max}\left(\mathcal{L}^*\mathcal{L} + \mathfrak{d}^*\mathfrak{d}\right) = 4p - 2$, completing the proof. ∎

## C.2 Proof of Theorem 3

**Proof** To prove Theorem 3, we first establish the boundedness of the sequence $\left\{\left(\boldsymbol{\Theta}^l, \boldsymbol{w}^l, \boldsymbol{Y}^l, \boldsymbol{y}^l\right)\right\}$ generated by Algorithm 1 in Lemma 10, and the monotonicity of $L_\rho\left(\boldsymbol{\Theta}^l, \boldsymbol{w}^l, \boldsymbol{Y}^l, \boldsymbol{y}^l\right)$ in Lemma 11.

**Lemma 10** *The sequence $\left\{ \left(\boldsymbol{\Theta}^l, \boldsymbol{w}^l, \boldsymbol{Y}^l, \boldsymbol{y}^l\right)\right\}$ generated by Algorithm 1 is bounded.*

**Proof** Let $\boldsymbol{w}^0$, $\boldsymbol{Y}^0$ and $\boldsymbol{y}^0$ be the initialization of the sequences $\{\boldsymbol{w}^l\}$, $\{\boldsymbol{Y}^l\}$ and $\{\boldsymbol{y}^l\}$, respectively, and $\left\|\boldsymbol{w}^0\right\|$, $\left\|\boldsymbol{Y}^0\right\|_{\mathrm{F}}$ and $\left\|\boldsymbol{y}^0\right\|$ are bounded. We prove the boundedness of the sequence by induction.

Recall that the sequence $\left\{\boldsymbol{\Theta}^l\right\}$ is established by

$$\boldsymbol{\Theta}^l = \frac{1}{2\rho}\boldsymbol{U}^{l-1}\left(\boldsymbol{\Gamma}^{l-1} + \sqrt{\left(\boldsymbol{\Gamma}^{l-1}\right)^2 + 4\rho\boldsymbol{I}}\right)\boldsymbol{U}^{l-1\top}, \tag{30}$$

where $\boldsymbol{\Gamma}^{l-1}$ contains the largest $p-k$ eigenvalues of $\rho\mathcal{L}\boldsymbol{w}^{l-1} - \boldsymbol{Y}^{l-1}$, and $\boldsymbol{U}^{l-1}$ contains the corresponding eigenvectors. When $l = 1$, $\left\|\boldsymbol{\Gamma}^0\right\|_{\mathrm{F}}$ is bounded since both $\left\|\boldsymbol{w}^0\right\|$ and $\left\|\boldsymbol{Y}^0\right\|_{\mathrm{F}}$ are bounded. Therefore, we can conclude that $\left\|\boldsymbol{\Theta}^1\right\|_{\mathrm{F}}$ is bounded.

The sequence $\left\{\boldsymbol{w}^l\right\}$ is established by solving the subproblems of the form

$$\min_{\boldsymbol{w}\geq\boldsymbol{0}} \frac{\rho}{2}\boldsymbol{w}^\top\left(\mathfrak{d}^*\mathfrak{d} + \mathcal{L}^*\mathcal{L}\right)\boldsymbol{w} - \left\langle \boldsymbol{w}, \mathcal{L}^*\left(\boldsymbol{Y}^{l-1} + \rho\boldsymbol{\Theta}^l\right) - \mathfrak{d}^*\left(\boldsymbol{y}^{l-1} - \rho\boldsymbol{d}\right)\right\rangle$$
$$+ \frac{p+\nu}{n}\sum_{i=1}^n \log\left(1 + \frac{\boldsymbol{x}_i^\top\mathcal{L}\boldsymbol{w}\boldsymbol{x}_i}{\nu}\right).$$

Let

$$g_l(\boldsymbol{w}) = \frac{\rho}{2}\boldsymbol{w}^\top\left(\mathfrak{d}^*\mathfrak{d} + \mathcal{L}^*\mathcal{L}\right)\boldsymbol{w} + \left\langle \boldsymbol{w}, \boldsymbol{a}^l\right\rangle + \frac{p+\nu}{n}\sum_{i=1}^n \log\left(1 + \frac{\boldsymbol{x}_i^\top\mathcal{L}\boldsymbol{w}\boldsymbol{x}_i}{\nu}\right), \tag{31}$$

where $\boldsymbol{a}^l = -\mathcal{L}^*\left(\boldsymbol{Y}^{l-1} + \rho\boldsymbol{\Theta}^l\right) + \mathfrak{d}^*\left(\boldsymbol{y}^{l-1} - \rho\boldsymbol{d}\right)$. Note that $\left\|\boldsymbol{a}^1\right\|$ is bounded because $\left\|\boldsymbol{Y}^0\right\|_{\mathrm{F}}$, $\left\|\boldsymbol{y}^0\right\|$, and $\left\|\boldsymbol{\Theta}^1\right\|_{\mathrm{F}}$ are bounded.

From Lemma 2, we known that $\mathcal{L}^*\mathcal{L}$ is a positive definite matrix with minimum eigenvalue $\lambda_{\min}\left(\mathcal{L}^*\mathcal{L}\right) = 2$. On the other hand, $\mathfrak{d}^*\mathfrak{d}$ is a positive semi-definite matrix as follows,

$$\lambda_{\min}\left(\mathfrak{d}^*\mathfrak{d}\right) = \sup_{\boldsymbol{x}\neq\boldsymbol{0}} \frac{\boldsymbol{x}^\top\mathfrak{d}^*\mathfrak{d}\boldsymbol{x}}{\boldsymbol{x}^\top\boldsymbol{x}} = \sup_{\boldsymbol{x}\neq\boldsymbol{0}} \frac{\left\langle\mathfrak{d}\boldsymbol{x}, \mathfrak{d}\boldsymbol{x}\right\rangle}{\boldsymbol{x}^\top\boldsymbol{x}} \geq 0. \tag{32}$$

Since $\log\left(1 + \frac{\boldsymbol{x}_i^\top\mathcal{L}\boldsymbol{w}\boldsymbol{x}_i}{\nu}\right) \geq 0$ for any $\boldsymbol{w} \geq \boldsymbol{0}$, we have

$$\lim_{\|\boldsymbol{w}\|\to+\infty} g_1(\boldsymbol{w}) \geq \lim_{\|\boldsymbol{w}\|\to+\infty} \rho\boldsymbol{w}^\top\boldsymbol{w} + \left\langle\boldsymbol{w}, \boldsymbol{a}^1\right\rangle = +\infty. \tag{33}$$

Thus $g_1(\boldsymbol{w})$ is coercive. Recall that we solve the optimization (31) by the MM framework. Hence, the objective function value is monotonically decreasing as a function of the iterations, and $\boldsymbol{w}^l$ is a stationary point of (31). Then the coercivity of $g_1(\boldsymbol{w})$ yields the boundedness of $\left\|\boldsymbol{w}^1\right\|$.

According to the dual variable updates, we obtain

$$\boldsymbol{Y}^1 = \boldsymbol{Y}^0 + \rho\left(\boldsymbol{\Theta}^1 - \mathcal{L}\boldsymbol{w}^1\right), \tag{34}$$

and

$$\boldsymbol{y}^1 = \boldsymbol{y}^0 + \rho\left(\mathfrak{d}\boldsymbol{w}^1 - \boldsymbol{d}\right). \tag{35}$$

It is obvious that both $\left\|\boldsymbol{Y}^1\right\|_{\mathrm{F}}$ and $\left\|\boldsymbol{y}^1\right\|$ are bounded. Therefore, it holds for $l = 1$ that $\left\{\left(\boldsymbol{\Theta}^l, \boldsymbol{w}^l, \boldsymbol{Y}^l, \boldsymbol{y}^l\right)\right\}$ is bounded.

Now we assume that $\left\{\left(\boldsymbol{\Theta}^{l-1}, \boldsymbol{w}^{l-1}, \boldsymbol{Y}^{l-1}, \boldsymbol{y}^{l-1}\right)\right\}$ is bounded for some $l \geq 1$, and check the boundedness of $\left\{\left(\boldsymbol{\Theta}^l, \boldsymbol{w}^l, \boldsymbol{Y}^l, \boldsymbol{y}^l\right)\right\}$. Similar to the proof in (30), we can prove that $\left\|\boldsymbol{\Theta}^l\right\|_{\mathrm{F}}$ is bounded. By (31), we can also obtain the boundedness of $\left\|\boldsymbol{w}^l\right\|$. We can also obtain that $\left\|\boldsymbol{Y}^l\right\|$ and $\|\boldsymbol{y}\|^l$ are bounded according to the boundedness of $\left\|\boldsymbol{\Theta}^l\right\|_{\mathrm{F}}$, $\left\|\boldsymbol{w}^l\right\|$, $\left\|\boldsymbol{Y}^{l-1}\right\|$ and $\left\|\boldsymbol{y}^{l-1}\right\|$. Thus, $\left\{\left(\boldsymbol{\Theta}^l, \boldsymbol{w}^l, \boldsymbol{Y}^l, \boldsymbol{y}^l\right)\right\}$ is bounded, completing the induction. Therefore, we establish the boundedness of the sequence $\left\{\left(\boldsymbol{\Theta}^l, \boldsymbol{w}^l, \boldsymbol{Y}^l, \boldsymbol{y}^l\right)\right\}$. ∎

**Lemma 11** *The sequence* $L_\rho\left(\mathbf{\Theta}^l, \boldsymbol{w}^l, \boldsymbol{Y}^l, \boldsymbol{y}^l\right)$ *generated by Algorithm 1 is lower bounded, and*

$$L_\rho\left(\mathbf{\Theta}^{l+1}, \boldsymbol{w}^{l+1}, \boldsymbol{Y}^{l+1}, \boldsymbol{y}^{l+1}\right) \le L_\rho\left(\mathbf{\Theta}^l, \boldsymbol{w}^l, \boldsymbol{Y}^l, \boldsymbol{y}^l\right), \quad \forall l \in \mathbb{N}_+, \tag{36}$$

*holds for any sufficiently large* $\rho$.

**Proof** According to Problem (1) in the manuscript, we have

$$L_\rho(\mathbf{\Theta}^l, \boldsymbol{w}^l, \boldsymbol{Y}^l, \boldsymbol{y}^l) = \frac{p+\nu}{n} \sum_{i=1}^n \log\left(1 + \frac{\boldsymbol{x}_i^\top \mathcal{L}\boldsymbol{w}^l \boldsymbol{x}_i}{\nu}\right) - \log\det\left(\mathbf{\Theta}^l + \boldsymbol{J}\right) + \langle \boldsymbol{y}^l, \mathfrak{d}\boldsymbol{w}^l - \boldsymbol{d}\rangle$$
$$+ \frac{\rho}{2}\left\|\mathfrak{d}\boldsymbol{w}^l - \boldsymbol{d}\right\|_2^2 + \langle \boldsymbol{Y}^l, \mathbf{\Theta}^l - \mathcal{L}\boldsymbol{w}^l\rangle + \frac{\rho}{2}\left\|\mathbf{\Theta}^l - \mathcal{L}\boldsymbol{w}^l\right\|_F^2. \tag{37}$$

We can see that the lower boundedness of the sequence $L_\rho\left(\mathbf{\Theta}^l, \boldsymbol{w}^l, \boldsymbol{Y}^l, \boldsymbol{y}^l\right)$ can be established by the boundedness of $\left\{\left(\mathbf{\Theta}^l, \boldsymbol{w}^l, \boldsymbol{Y}^l, \boldsymbol{y}^l\right)\right\}$ in Lemma 10.

We first establish that

$$L_\rho\left(\mathbf{\Theta}^{l+1}, \boldsymbol{w}^l, \boldsymbol{Y}^l, \boldsymbol{y}^l\right) \le L_\rho\left(\mathbf{\Theta}^l, \boldsymbol{w}^l, \boldsymbol{Y}^l, \boldsymbol{y}^l\right), \quad \forall l \in \mathbb{N}_+. \tag{38}$$

One has

$$L_\rho(\mathbf{\Theta}^{l+1}, \boldsymbol{w}^l, \boldsymbol{Y}^l, \boldsymbol{y}^l) - L_\rho(\mathbf{\Theta}^{l+1}, \boldsymbol{w}^{l+1}, \boldsymbol{Y}^{l+1}, \boldsymbol{y}^{l+1})$$
$$= \underbrace{\langle \boldsymbol{y}^l, \mathfrak{d}\boldsymbol{w}^l - \boldsymbol{d}\rangle - \langle \boldsymbol{y}^{l+1}, \mathfrak{d}\boldsymbol{w}^{l+1} - \boldsymbol{d}\rangle}_{I_1} + \underbrace{\langle \boldsymbol{Y}^l, \mathbf{\Theta}^{l+1} - \mathcal{L}\boldsymbol{w}^l\rangle - \langle \boldsymbol{Y}^{l+1}, \mathbf{\Theta}^{l+1} - \mathcal{L}\boldsymbol{w}^{l+1}\rangle}_{I_2}$$
$$+ r\left(\mathcal{L}\boldsymbol{w}^l\right) - r\left(\mathcal{L}\boldsymbol{w}^{l+1}\right) + \frac{\rho}{2}\left\|\mathfrak{d}\boldsymbol{w}^l - \boldsymbol{d}\right\|_2^2 - \frac{\rho}{2}\left\|\mathfrak{d}\boldsymbol{w}^{l+1} - \boldsymbol{d}\right\|_2^2$$
$$+ \frac{\rho}{2}\left\|\mathbf{\Theta}^{l+1} - \mathcal{L}\boldsymbol{w}^l\right\|_F^2 - \frac{\rho}{2}\left\|\mathbf{\Theta}^{l+1} - \mathcal{L}\boldsymbol{w}^{l+1}\right\|_F^2, \tag{39}$$

where $r(\boldsymbol{L}) = \frac{p+\nu}{n}\sum_{i=1}^n \log\left(1 + \frac{\boldsymbol{x}_i^\top \boldsymbol{L}\boldsymbol{x}_i}{\nu}\right)$.

From the dual variables updates, it is easy to see that

$$I_1 + I_2 = \left\langle \mathfrak{d}^*\boldsymbol{y}^l - \mathcal{L}^*\boldsymbol{Y}^l, \boldsymbol{w}^l - \boldsymbol{w}^{l+1}\right\rangle - \rho\left\|\mathfrak{d}\boldsymbol{w}^{l+1} - \boldsymbol{d}\right\|_2^2 - \rho\left\|\mathbf{\Theta}^{l+1} - \mathcal{L}\boldsymbol{w}^{l+1}\right\|_F^2. \tag{40}$$

According to the convergence result of the majorization-minimization framework [35, 36], $\boldsymbol{w}^{l+1}$ is a stationary point of the following problem

$$\underset{\boldsymbol{w}\ge\boldsymbol{0}}{\text{minimize}}\; r(\mathcal{L}\boldsymbol{w}) + \frac{\rho}{2}\boldsymbol{w}^\top\left(\mathfrak{d}^*\mathfrak{d} + \mathcal{L}^*\mathcal{L}\right)\boldsymbol{w} - \left\langle \boldsymbol{w}, \mathcal{L}^*\left(\boldsymbol{Y}^l + \rho\mathbf{\Theta}^{l+1}\right) - \mathfrak{d}^*\left(\boldsymbol{y}^l - \rho\boldsymbol{d}\right)\right\rangle. \tag{41}$$

The set of the stationary points for the optimization (41) is defined by

$$\mathcal{X} = \left\{\boldsymbol{w} \mid \nabla g_l(\boldsymbol{w})^\top(\boldsymbol{z} - \boldsymbol{w}) \ge 0,\; \forall \boldsymbol{z} \ge \boldsymbol{0}\right\}, \tag{42}$$

where $g_l(\boldsymbol{w})$ is the objective function in (41). The existence of the limit point can be guaranteed by the the coercivity of $g_l(\boldsymbol{w})$, which has been established in the proof of Lemma 10. By taking $\boldsymbol{z} = \boldsymbol{w}^l$ and $\boldsymbol{w} = \boldsymbol{w}^{l+1}$ in (42), we obtain

$$\left(\mathcal{L}^*\left(\nabla r\left(\mathcal{L}\boldsymbol{w}^{l+1}\right)\right) + \rho\left(\mathfrak{d}^*\mathfrak{d} + \mathcal{L}^*\mathcal{L}\right)\boldsymbol{w}^{l+1} - \mathcal{L}^*\left(\boldsymbol{Y}^l + \rho\mathbf{\Theta}^{l+1}\right) + \mathfrak{d}^*\left(\boldsymbol{y}^l - \rho\boldsymbol{d}\right)\right)^\top\left(\boldsymbol{w}^l - \boldsymbol{w}^{l+1}\right) \ge 0.$$

Thus, we have

$$\langle \mathfrak{d}^*\boldsymbol{y}^l - \mathcal{L}^*\boldsymbol{Y}^l, \boldsymbol{w}^l - \boldsymbol{w}^{l+1}\rangle \ge -\left\langle \nabla r\left(\mathcal{L}\boldsymbol{w}^{l+1}\right), \mathcal{L}\boldsymbol{w}^l - \mathcal{L}\boldsymbol{w}^{l+1}\right\rangle$$
$$+ \rho\left\langle -\left(\mathfrak{d}^*\mathfrak{d} + \mathcal{L}^*\mathcal{L}\right)\boldsymbol{w}^{l+1} + \mathcal{L}^*\mathbf{\Theta}^{l+1} + \mathfrak{d}^*\boldsymbol{d}, \boldsymbol{w}^l - \boldsymbol{w}^{l+1}\right\rangle, \tag{43}$$

Plugging (40) and (43) into (39), we obtain

$$L_\rho(\mathbf{\Theta}^{l+1}, \boldsymbol{w}^l, \boldsymbol{Y}^l, \boldsymbol{y}^l) - L_\rho(\mathbf{\Theta}^{l+1}, \boldsymbol{w}^{l+1}, \boldsymbol{Y}^{l+1}, \boldsymbol{y}^{l+1})$$
$$\ge \frac{\rho}{2}\left\|\mathfrak{d}\boldsymbol{w}^{l+1} - \mathfrak{d}\boldsymbol{w}^l\right\|_2^2 + \frac{\rho}{2}\left\|\mathcal{L}\boldsymbol{w}^{l+1} - \mathcal{L}\boldsymbol{w}^l\right\|_2^2 - \rho\left\|\mathfrak{d}\boldsymbol{w}^{l+1} - \boldsymbol{d}\right\|_2^2 - \rho\left\|\mathcal{L}\boldsymbol{w}^{l+1} - \mathbf{\Theta}^{l+1}\right\|_F^2$$
$$+ r\left(\mathcal{L}\boldsymbol{w}^l\right) - r\left(\mathcal{L}\boldsymbol{w}^{l+1}\right) - \left\langle \nabla r\left(\mathcal{L}\boldsymbol{w}^{l+1}\right), \mathcal{L}\boldsymbol{w}^l - \mathcal{L}\boldsymbol{w}^{l+1}\right\rangle$$
$$\ge \frac{\rho}{2}\left\|\mathfrak{d}\boldsymbol{w}^{l+1} - \mathfrak{d}\boldsymbol{w}^l\right\|_2^2 + \frac{\rho - L_r}{2}\left\|\mathcal{L}\boldsymbol{w}^{l+1} - \mathcal{L}\boldsymbol{w}^l\right\|_F^2 - \frac{1}{\rho}\left\|\boldsymbol{y}^{l+1} - \boldsymbol{y}^l\right\|_2^2 - \frac{1}{\rho}\left\|\boldsymbol{Y}^{l+1} - \boldsymbol{Y}^l\right\|_F^2, \tag{44}$$

where the last inequality is due to the fact that $r(\boldsymbol{L})$ is a concave function and has $L_r$-Lipschitz continuous gradient, in which $L_r > 0$ is a constant, thus we have

$$r\left(\mathcal{L}\boldsymbol{w}^l\right) - r\left(\mathcal{L}\boldsymbol{w}^{l+1}\right) - \left\langle \nabla r\left(\mathcal{L}\boldsymbol{w}^{l+1}\right), \mathcal{L}\boldsymbol{w}^l - \mathcal{L}\boldsymbol{w}^{l+1}\right\rangle \geq -\frac{L_r}{2}\left\|\mathcal{L}\boldsymbol{w}^{l+1} - \mathcal{L}\boldsymbol{w}^l\right\|_{\mathrm{F}}^2. \quad (45)$$

By calculation, we obtain that if $\rho$ is sufficiently large such that

$$\rho \geq \max\left(2L_r,\ \max_l \frac{C\left(\left\|\boldsymbol{Y}^{l+1} - \boldsymbol{Y}^l\right\|_{\mathrm{F}}^2 + \left\|\boldsymbol{y}^{l+1} - \boldsymbol{y}^l\right\|_2^2\right)^{\frac{1}{2}}}{\left(\left\|\mathcal{L}\boldsymbol{w}^{l+1} - \mathcal{L}\boldsymbol{w}^l\right\|_{\mathrm{F}}^2 + \left\|\mathfrak{d}\boldsymbol{w}^{l+1} - \mathfrak{d}\boldsymbol{w}^l\right\|_2^2\right)^{\frac{1}{2}}}\right), \quad (46)$$

holds with some constant $C > 2$, then, together with (38), we conclude that

$$L_\rho(\boldsymbol{\Theta}^l, \boldsymbol{w}^l, \boldsymbol{Y}^l, \boldsymbol{y}^l) \geq L_\rho(\boldsymbol{\Theta}^{l+1}, \boldsymbol{w}^l, \boldsymbol{Y}^l, \boldsymbol{y}^l) \geq L_\rho(\boldsymbol{\Theta}^{l+1}, \boldsymbol{w}^{l+1}, \boldsymbol{Y}^{l+1}, \boldsymbol{y}^{l+1}), \quad (47)$$

for any $l \in \mathbb{N}_+$. ∎

Now we are ready to prove Theorem 3. By Lemma 10, the sequence $\{(\boldsymbol{\Theta}^l, \boldsymbol{w}^l, \boldsymbol{Y}^l, \boldsymbol{y}^l)\}$ generated by Algorithm 1 is bounded. Therefore, there exists at least one convergent subsequence $\{(\boldsymbol{\Theta}^{l_s}, \boldsymbol{w}^{l_s}, \boldsymbol{Y}^{l_s}, \boldsymbol{y}^{l_s})\}_{s\in\mathbb{N}}$, which converges to the limit point denoted by $\{(\boldsymbol{\Theta}^{l_\infty}, \boldsymbol{w}^{l_\infty}, \boldsymbol{Y}^{l_\infty}, \boldsymbol{y}^{l_\infty})\}$. By Lemma 11, we obtain that the sequence $L_\rho\left(\boldsymbol{\Theta}^l, \boldsymbol{w}^l, \boldsymbol{Y}^l, \boldsymbol{y}^l\right)$ defined in (37) is monotonically decreasing and lower bounded, implying that $L_\rho\left(\boldsymbol{\Theta}^l, \boldsymbol{w}^l, \boldsymbol{Y}^l, \boldsymbol{y}^l\right)$ is convergent.

We can then obtain $\lim_{l\to+\infty} L_\rho\left(\boldsymbol{\Theta}^l, \boldsymbol{w}^l, \boldsymbol{Y}^l, \boldsymbol{y}^l\right) = L_\rho\left(\boldsymbol{\Theta}^{l_\infty}, \boldsymbol{w}^{l_\infty}, \boldsymbol{Y}^{l_\infty}, \boldsymbol{y}^{l_\infty}\right)$. Then, (44), (46), and (47) together yield

$$L_\rho(\boldsymbol{\Theta}^l, \boldsymbol{w}^l, \boldsymbol{Y}^l, \boldsymbol{y}^l) - L_\rho(\boldsymbol{\Theta}^{l+1}, \boldsymbol{w}^{l+1}, \boldsymbol{Y}^{l+1}, \boldsymbol{y}^{l+1})$$
$$\geq \frac{C^2 - 4}{4}\rho\left(\left\|\mathcal{L}\boldsymbol{w}^{l+1} - \boldsymbol{\Theta}^{l+1}\right\|_{\mathrm{F}}^2 + \left\|\mathfrak{d}\boldsymbol{w}^{l+1} - \boldsymbol{d}\right\|_2^2\right). \quad (48)$$

The convergence of $L_\rho\left(\boldsymbol{\Theta}^l, \boldsymbol{w}^l, \boldsymbol{Y}^l, \boldsymbol{y}^l\right)$ yields

$$\lim_{l\to+\infty}\left\|\mathcal{L}\boldsymbol{w}^l - \boldsymbol{\Theta}^l\right\|_{\mathrm{F}} = 0, \quad \text{and} \quad \lim_{l\to+\infty}\left\|\mathfrak{d}\boldsymbol{w}^l - \boldsymbol{d}\right\|_2 = 0. \quad (49)$$

By the updating of $\boldsymbol{Y}^{l+1}$ and $\boldsymbol{y}^{l+1}$, we can get

$$\lim_{l\to+\infty}\left\|\boldsymbol{Y}^{l+1} - \boldsymbol{Y}^l\right\|_{\mathrm{F}} = 0, \quad \text{and} \quad \lim_{l\to+\infty}\left\|\boldsymbol{y}^{l+1} - \boldsymbol{y}^l\right\|_2 = 0. \quad (50)$$

Together with (44), we obtain

$$\lim_{l\to+\infty}\left\|\mathfrak{d}\boldsymbol{w}^{l+1} - \mathfrak{d}\boldsymbol{w}^l\right\|_2 = 0 \quad \text{and} \quad \lim_{l\to+\infty}\left\|\mathcal{L}\boldsymbol{w}^{l+1} - \mathcal{L}\boldsymbol{w}^l\right\|_{\mathrm{F}} = 0. \quad (51)$$

For the limit point $\{(\boldsymbol{\Theta}^{l_\infty}, \boldsymbol{w}^{l_\infty}, \boldsymbol{Y}^{l_\infty}, \boldsymbol{y}^{l_\infty})\}$ of any subsequence $\{(\boldsymbol{\Theta}^{l_s}, \boldsymbol{w}^{l_s}, \boldsymbol{Y}^{l_s}, \boldsymbol{y}^{l_s})\}_{s\in\mathbb{N}}$, $\boldsymbol{\Theta}^{l_\infty}$ minimizes the following subproblem

$$\boldsymbol{\Theta}^{l_\infty} = \underset{\boldsymbol{\Theta}\succeq\boldsymbol{0}}{\arg\min}\ -\log\det(\boldsymbol{\Theta} + \boldsymbol{J}) + \left\langle\boldsymbol{\Theta}, \boldsymbol{Y}^{l_\infty-1}\right\rangle + \frac{\rho}{2}\left\|\boldsymbol{\Theta} - \mathcal{L}\boldsymbol{w}^{l_\infty-1}\right\|_{\mathrm{F}}^2$$
$$= \underset{\boldsymbol{\Theta}\succeq\boldsymbol{0}}{\arg\min}\ -\log\det(\boldsymbol{\Theta} + \boldsymbol{J}) + \left\langle\boldsymbol{\Theta}, \boldsymbol{Y}^{l_\infty}\right\rangle + \frac{\rho}{2}\left\|\boldsymbol{\Theta} - \mathcal{L}\boldsymbol{w}^{l_\infty}\right\|_{\mathrm{F}}^2$$
$$- \left\langle\boldsymbol{\Theta}, \boldsymbol{Y}^{l_\infty} - \boldsymbol{Y}^{l_\infty-1}\right\rangle + \rho\left\langle\boldsymbol{\Theta}, \mathcal{L}\boldsymbol{w}^{l_\infty} - \mathcal{L}\boldsymbol{w}^{l_\infty-1}\right\rangle.$$

By (50) and (51), we conclude that $\boldsymbol{\Theta}^{l_\infty}$ satisfies the condition of stationary point of $L_\rho(\boldsymbol{\Theta}, \boldsymbol{w}, \boldsymbol{V}, \boldsymbol{Y}, \boldsymbol{y})$ with respect to $\boldsymbol{\Theta}$. Similarly, from the convergence results of the MM framework, we know that $\boldsymbol{w}^{l_\infty}$ is a stationary point of the subproblem

$$\underset{\boldsymbol{w}\geq\boldsymbol{0}}{\text{minimize}}\ \frac{\rho}{2}\boldsymbol{w}^\top\left(\mathfrak{d}^*\mathfrak{d} + \mathcal{L}^*\mathcal{L}\right)\boldsymbol{w} + \left\langle\boldsymbol{w}, \mathfrak{d}^*\left(\boldsymbol{y}^{l_\infty} - \rho\boldsymbol{d}\right)\right\rangle - \left\langle\boldsymbol{w}, \mathcal{L}^*\left(\boldsymbol{Y}^{l_\infty} + \rho\boldsymbol{\Theta}^{l_\infty}\right)\right\rangle$$
$$+ \left\langle\mathcal{L}\boldsymbol{w}, \boldsymbol{Y}^{l_\infty} - \boldsymbol{Y}^{l_\infty-1}\right\rangle + \left\langle\mathfrak{d}\boldsymbol{w}, \boldsymbol{y}^{l_\infty-1} - \boldsymbol{y}^{l_\infty}\right\rangle + r(\mathcal{L}\boldsymbol{w}).$$

By (50) and (51), $\boldsymbol{w}^{l\infty}$ satisfies the condition of stationary point of $L_\rho(\boldsymbol{\Theta}, \boldsymbol{w}, \boldsymbol{Y}, \boldsymbol{y})$ with respect to $\boldsymbol{w}$.

To sum up, we can conclude that any limit point $\left\{\left(\boldsymbol{\Theta}^{l\infty}, \boldsymbol{w}^{l\infty}, \boldsymbol{Y}^{l\infty}, \boldsymbol{y}^{l\infty}\right)\right\}$ of the sequence generated by Algorithm 1 is a stationary point of $L_\rho(\boldsymbol{\Theta}, \boldsymbol{w}, \boldsymbol{Y}, \boldsymbol{y})$. ∎

## C.3  Proof of Theorem 4

**Proof**  Similarly to the proof conducted for Theorem 3, to prove Theorem 4, we first establish the boundedness of the sequence $\left\{\left(\boldsymbol{\Theta}^l, \boldsymbol{w}^l, \boldsymbol{V}^l, \boldsymbol{Y}^l, \boldsymbol{y}^l\right)\right\}$ generated by Algorithm 2 in Lemma 12, and the monotonicity of $L_\rho\left(\boldsymbol{\Theta}^l, \boldsymbol{w}^l, \boldsymbol{V}^l, \boldsymbol{Y}^l, \boldsymbol{y}^l\right)$ in Lemma 13.

**Lemma 12**  *The sequence $\left\{\left(\boldsymbol{\Theta}^l, \boldsymbol{w}^l, \boldsymbol{V}^l, \boldsymbol{Y}^l, \boldsymbol{y}^l\right)\right\}$ generated by Algorithm 2 is bounded.*

**Proof**  Let $\boldsymbol{w}^0$, $\boldsymbol{V}^0$, $\boldsymbol{Y}^0$ and $\boldsymbol{y}^0$ be the initialization of the sequences $\{\boldsymbol{w}^l\}$, $\{\boldsymbol{V}^l\}$, $\{\boldsymbol{Y}^l\}$ and $\{\boldsymbol{y}^l\}$, respectively, and $\left\|\boldsymbol{w}^0\right\|$, $\left\|\boldsymbol{V}^0\right\|_{\mathrm{F}}$, $\left\|\boldsymbol{Y}^0\right\|_{\mathrm{F}}$ and $\left\|\boldsymbol{y}^0\right\|$ are bounded.

We prove the lemma by induction. Recall that the sequence $\left\{\boldsymbol{\Theta}^l\right\}$ is established by

$$\boldsymbol{\Theta}^l = \frac{1}{2\rho}\boldsymbol{U}^{l-1}\left(\boldsymbol{\Gamma}^{l-1} + \sqrt{\left(\boldsymbol{\Gamma}^{l-1}\right)^2 + 4\rho\boldsymbol{I}}\right)\boldsymbol{U}^{l-1\top}, \tag{52}$$

where $\boldsymbol{\Gamma}^{l-1}$ contains the largest $p - k$ eigenvalues of $\rho\mathcal{L}\boldsymbol{w}^{l-1} - \boldsymbol{Y}^{l-1}$, and $\boldsymbol{U}^{l-1}$ contains the corresponding eigenvectors. When $l = 1$, $\left\|\boldsymbol{\Gamma}^0\right\|_{\mathrm{F}}$ is bounded since both $\left\|\boldsymbol{w}^0\right\|$ and $\left\|\boldsymbol{Y}^0\right\|_{\mathrm{F}}$ are bounded. Therefore, we can conclude that $\left\|\boldsymbol{\Theta}^1\right\|_{\mathrm{F}}$ is bounded.

The sequence $\left\{\boldsymbol{w}^l\right\}$ is established by solving the subproblems that tantamount to (31) except for the additional linear term $\eta\langle\boldsymbol{w}, \mathcal{L}^*\left(\boldsymbol{V}^l\boldsymbol{V}^{l\top}\right)\rangle$. Hence, $g_1(\boldsymbol{w}) + \eta\langle\boldsymbol{w}, \mathcal{L}^*\left(\boldsymbol{V}^1\boldsymbol{V}^{1\top}\right)\rangle$ is still coercive, which implies that $\left\|\boldsymbol{w}^1\right\|$ is bounded.

The feasible set of $\boldsymbol{V}$ is the set of ordered orthonormal $k$-frames in $\mathbb{R}^p$, thus $\left\|\boldsymbol{V}^l\right\|_{\mathrm{F}}$ is bounded for any $l \geq 1$.

From the updates of $\boldsymbol{Y}$ and $\boldsymbol{y}$, it is obvious that both $\left\|\boldsymbol{Y}^1\right\|_{\mathrm{F}}$ and $\left\|\boldsymbol{y}^1\right\|$ are bounded. Therefore, it holds for $l = 1$ that $\left\{\left(\boldsymbol{\Theta}^l, \boldsymbol{w}^l, \boldsymbol{V}^l, \boldsymbol{Y}^l, \boldsymbol{y}^l\right)\right\}$ is bounded.

Assume that $\left\{\left(\boldsymbol{\Theta}^{l-1}, \boldsymbol{w}^{l-1}, \boldsymbol{V}^{l-1}, \boldsymbol{Y}^{l-1}, \boldsymbol{y}^{l-1}\right)\right\}$ is bounded for some $l \geq 1$, *i.e.*, each term in $\left\{\left(\boldsymbol{\Theta}^{l-1}, \boldsymbol{w}^{l-1}, \boldsymbol{V}^{l-1}, \boldsymbol{Y}^{l-1}, \boldsymbol{y}^{l-1}\right)\right\}$ is bounded under $\ell_2$-norm or Frobenius norm. Following (52), we can obtain that $\left\|\boldsymbol{\Theta}^l\right\|_{\mathrm{F}}$ is bounded.

Due to the coercivity of $g_l(\boldsymbol{w}) + \eta\langle\boldsymbol{w}, \mathcal{L}^*\left(\boldsymbol{V}^l\boldsymbol{V}^{l\top}\right)\rangle$, we can get that $\left\|\boldsymbol{w}^l\right\|$ is bounded. It is also obvious that $\left\|\boldsymbol{Y}^l\right\|$ and $\left\|\boldsymbol{y}\right\|^l$ are bounded, because of the boundedness of $\left\|\boldsymbol{\Theta}^l\right\|_{\mathrm{F}}$, $\left\|\boldsymbol{w}^l\right\|$, $\left\|\boldsymbol{Y}^{l-1}\right\|$ and $\left\|\boldsymbol{y}^{l-1}\right\|$. Thus, $\left\{\left(\boldsymbol{\Theta}^l, \boldsymbol{w}^l, \boldsymbol{V}^l, \boldsymbol{Y}^l, \boldsymbol{y}^l\right)\right\}$ is bounded, completing the induction. Therefore, we can conclude that the sequence $\left\{\left(\boldsymbol{\Theta}^l, \boldsymbol{w}^l, \boldsymbol{V}^l, \boldsymbol{Y}^l, \boldsymbol{y}^l\right)\right\}$ is bounded. ∎

**Lemma 13**  *The sequence $L_\rho\left(\boldsymbol{\Theta}^l, \boldsymbol{w}^l, \boldsymbol{V}^l, \boldsymbol{Y}^l, \boldsymbol{y}^l\right)$ generated by Algorithm 2 is lower bounded, and*

$$L_\rho\left(\boldsymbol{\Theta}^{l+1}, \boldsymbol{w}^{l+1}, \boldsymbol{V}^{l+1}, \boldsymbol{Y}^{l+1}, \boldsymbol{y}^{l+1}\right) \leq L_\rho\left(\boldsymbol{\Theta}^l, \boldsymbol{w}^l, \boldsymbol{V}^l, \boldsymbol{Y}^l, \boldsymbol{y}^l\right), \quad \forall l \in \mathbb{N}_+, \tag{53}$$

*holds for any sufficiently large $\rho$.*

**Proof**  According to Problem (16) in the manuscript, we have

$$
\begin{aligned}
L_\rho(\boldsymbol{\Theta}^l, \boldsymbol{w}^l, \boldsymbol{V}^l, \boldsymbol{Y}^l, \boldsymbol{y}^l) = {}& \frac{p+\nu}{n}\sum_{i=1}^n \log\left(1 + \frac{\boldsymbol{x}_i^\top \mathcal{L}\boldsymbol{w}^l\boldsymbol{x}_i}{\nu}\right) + \operatorname{tr}\left(\mathcal{L}\boldsymbol{w}^l\left(\eta\boldsymbol{V}^l\left(\boldsymbol{V}^l\right)^\top\right)\right) \\
& - \log\det{}^*\left(\boldsymbol{\Theta}^l\right) + \left\langle\boldsymbol{y}^l, \mathfrak{d}\boldsymbol{w}^l - \boldsymbol{d}\right\rangle + \frac{\rho}{2}\left\|\mathfrak{d}\boldsymbol{w}^l - \boldsymbol{d}\right\|_2^2 \\
& + \left\langle\boldsymbol{Y}^l, \boldsymbol{\Theta}^l - \mathcal{L}\boldsymbol{w}^l\right\rangle + \frac{\rho}{2}\left\|\boldsymbol{\Theta}^l - \mathcal{L}\boldsymbol{w}^l\right\|_{\mathrm{F}}^2.
\end{aligned}
\tag{54}
$$

We can see that the lower boundedness of the sequence $L_\rho\left(\Theta^l, w^l, V^l, Y^l, y^l\right)$ can be established by the boundedness of $\left\{\left(\Theta^l, w^l, V^l, Y^l, y^l\right)\right\}$ in Lemma 12.

We first establish that

$$L_\rho\left(\Theta^{l+1}, w^l, V^l, Y^l, y^l\right) \leq L_\rho\left(\Theta^l, w^l, V^l, Y^l, y^l\right), \quad \forall l \in \mathbb{N}_+. \tag{55}$$

We have

$$L_\rho(\Theta^{l+1}, w^l, V^l, Y^l, y^l) = \frac{p+\nu}{n}\sum_{i=1}^n \log\left(1 + \frac{x_i^\top \mathcal{L}w^l x_i}{\nu}\right) + \mathrm{tr}\left(\mathcal{L}w^l\left(\eta V^l\left(V^l\right)^\top\right)\right)$$
$$- \log\det{}^*\left(\Theta^{l+1}\right) + \langle y^l, \eth w^l - d\rangle + \frac{\rho}{2}\left\|\eth w^l - d\right\|_2^2$$
$$+ \langle Y^l, \Theta^{l+1} - \mathcal{L}w^l\rangle + \frac{\rho}{2}\left\|\Theta^{l+1} - \mathcal{L}w^l\right\|_\mathrm{F}^2.$$

Then we obtain

$$L_\rho(\Theta^{l+1}, w^l, V^l, Y^l, y^l) - L_\rho(\Theta^l, w^l, V^l, Y^l, y^l) = -\log\det{}^*\left(\Theta^{l+1}\right) + \langle Y^l, \Theta^{l+1}\rangle$$
$$+ \frac{\rho}{2}\left\|\Theta^{l+1} - \mathcal{L}w^l\right\|_\mathrm{F}^2 - \left(-\log\det{}^*\left(\Theta^l\right) + \langle Y^l, \Theta^l\rangle + \frac{\rho}{2}\left\|\Theta^l - \mathcal{L}w^l\right\|_\mathrm{F}^2\right).$$

Note that $\Theta^{l+1}$ minimizes the objective function

$$\Theta^{l+1} = \underset{\substack{\mathrm{rank}(\Theta)=p-k \\ \Theta \succeq 0}}{\arg\min} -\log\det{}^*(\Theta) + \langle \Theta, Y^l\rangle + \frac{\rho}{2}\left\|\Theta - \mathcal{L}w^l\right\|_\mathrm{F}^2. \tag{56}$$

Therefore

$$L_\rho(\Theta^{l+1}, w^l, V^l, Y^l, y^l) - L_\rho(\Theta^l, w^l, V^l, Y^l, y^l) \leq 0 \tag{57}$$

holds for any $l \in \mathbb{N}_+$.

One has

$$L_\rho(\Theta^{l+1}, w^l, V^l, Y^l, y^l) - L_\rho(\Theta^{l+1}, w^{l+1}, V^{l+1}, Y^{l+1}, y^{l+1})$$
$$= r(\mathcal{L}w^l) - r(\mathcal{L}w^{l+1}) + \underbrace{\mathrm{tr}\left(\eta\mathcal{L}w^l\left(V^l\left(V^l\right)^\top\right)\right) - \mathrm{tr}\left(\eta\mathcal{L}w^{l+1}\left(V^{l+1}\left(V^{l+1}\right)^\top\right)\right)}_{I_1}$$
$$+ \underbrace{\langle y^l, \eth w^l - d\rangle - \langle y^{l+1}, \eth w^{l+1} - d\rangle}_{I_2} + \underbrace{\langle Y^l, \Theta^{l+1} - \mathcal{L}w^l\rangle - \langle Y^{l+1}, \Theta^{l+1} - \mathcal{L}w^{l+1}\rangle}_{I_3}$$
$$+ \frac{\rho}{2}\left\|\Theta^{l+1} - \mathcal{L}w^l\right\|_\mathrm{F}^2 - \frac{\rho}{2}\left\|\Theta^{l+1} - \mathcal{L}w^{l+1}\right\|_\mathrm{F}^2 + \frac{\rho}{2}\left\|\eth w^l - d\right\|_2^2 - \frac{\rho}{2}\left\|\eth w^{l+1} - d\right\|_2^2, \tag{58}$$

where $r(L) = \frac{p+\nu}{n}\sum_{i=1}^n \log\left(1 + \frac{x_i^\top L x_i}{\nu}\right)$. The term $I_1$ can be written as

$$I_1 = \mathrm{tr}\left(\eta\mathcal{L}w^l\left(V^l\left(V^l\right)^\top\right)\right) - \mathrm{tr}\left(\eta\mathcal{L}w^{l+1}\left(V^l\left(V^l\right)^\top\right)\right)$$
$$+ \underbrace{\mathrm{tr}\left(\eta\mathcal{L}w^{l+1}\left(V^l\left(V^l\right)^\top\right)\right) - \mathrm{tr}\left(\eta\mathcal{L}w^{l+1}\left(V^{l+1}\left(V^{l+1}\right)^\top\right)\right)}_{I_{1a}}.$$

Note that $V^{l+1}$ is the optimal solution of the problem

$$\min_{V \in \mathbb{R}^{p \times k}} \mathrm{tr}\left(V^\top \mathcal{L}w^{l+1} V\right), \quad \text{subject to } V^\top V = I. \tag{59}$$

Thus the term $I_{1a} \geq 0$, and we can obtain

$$I_1 \geq \mathrm{tr}\left(\eta\mathcal{L}w^l\left(V^l\left(V^l\right)^\top\right)\right) - \mathrm{tr}\left(\eta\mathcal{L}w^{l+1}\left(V^l\left(V^l\right)^\top\right)\right). \tag{60}$$

For the term $I_2$, we have

$$I_2 = \langle y^l, \eth w^l - d\rangle - \langle y^l, \eth w^{l+1} - d\rangle - \rho\langle \eth w^{l+1} - d, \eth w^{l+1} - d\rangle$$
$$= \langle \eth^* y^l, w^l - w^{l+1}\rangle - \rho\left\|\eth w^{l+1} - d\right\|_2^2, \tag{61}$$

where the first equality is due to the updating of $y^{l+1}$ as below

$$y^{l+1} = y^l + \rho \left( \mathfrak{d} w^{l+1} - d \right). \tag{62}$$

For the term $I_3$, similarly, we have

$$
\begin{aligned}
I_3 &= \left\langle \boldsymbol{Y}^l, \boldsymbol{\Theta}^{l+1} - \mathcal{L} w^l \right\rangle - \left\langle \boldsymbol{Y}^l, \boldsymbol{\Theta}^{l+1} - \mathcal{L} w^{l+1} \right\rangle - \rho \left\langle \boldsymbol{\Theta}^{l+1} - \mathcal{L} w^{l+1}, \boldsymbol{\Theta}^{l+1} - \mathcal{L} w^{l+1} \right\rangle \\
&= \left\langle \mathcal{L}^* \boldsymbol{Y}^l, w^{l+1} - w^l \right\rangle - \rho \left\| \boldsymbol{\Theta}^{l+1} - \mathcal{L} w^{l+1} \right\|_{\mathrm{F}}^2,
\end{aligned}
\tag{63}
$$

where the first equality follows from

$$\boldsymbol{Y}^{l+1} = \boldsymbol{Y}^l + \rho \left( \boldsymbol{\Theta}^{l+1} - \mathcal{L} w^{l+1} \right). \tag{64}$$

Therefore, we can obtain

$$I_2 + I_3 = \left\langle \mathfrak{d}^* y^l - \mathcal{L}^* \boldsymbol{Y}^l, w^l - w^{l+1} \right\rangle - \rho \left\| \mathfrak{d} w^{l+1} - d \right\|_2^2 - \rho \left\| \boldsymbol{\Theta}^{l+1} - \mathcal{L} w^{l+1} \right\|_{\mathrm{F}}^2. \tag{65}$$

Recall that $w^{l+1}$ is a stationary point of the problem

$$
\begin{aligned}
\operatorname*{minimize}_{w \geq 0} \ &r(\mathcal{L} w) + \frac{\rho}{2} w^\top \left( \mathfrak{d}^* \mathfrak{d} + \mathcal{L}^* \mathcal{L} \right) w \\
&- \left\langle w, \mathcal{L}^* \left( \boldsymbol{Y}^l + \rho \boldsymbol{\Theta}^{l+1} - \eta \boldsymbol{V}^l \boldsymbol{V}^{l\top} \right) - \mathfrak{d}^* \left( y^l - \rho d \right) \right\rangle.
\end{aligned}
\tag{66}
$$

The set of the stationary points for the optimization (66) is defined by

$$\mathcal{X} = \left\{ w \mid \nabla q_l(w)^\top (z - w) \geq 0, \ \forall z \geq \mathbf{0} \right\}, \tag{67}$$

where $q_l(w)$ is the objective function in (66). By taking $z = w^l$ and $w = w^{l+1}$ in (67), we obtain

$$
\begin{aligned}
\left\langle \mathfrak{d}^* y^l - \mathcal{L}^* \boldsymbol{Y}^l, w^l - w^{l+1} \right\rangle &\geq - \left\langle \nabla r \left( \mathcal{L} w^{l+1} \right), \mathcal{L} w^l - \mathcal{L} w^{l+1} \right\rangle \\
&+ \rho \left\langle -\left( \mathfrak{d}^* \mathfrak{d} + \mathcal{L}^* \mathcal{L} \right) w^{l+1} + \mathcal{L}^* \boldsymbol{\Theta}^{l+1} + \mathfrak{d}^* d - \frac{\eta}{\rho} \boldsymbol{V}^l \boldsymbol{V}^{l\top}, w^l - w^{l+1} \right\rangle,
\end{aligned}
\tag{68}
$$

Substituting (65) and (68) into (58), we obtain

$$
\begin{aligned}
&L_\rho(\boldsymbol{\Theta}^{l+1}, w^l, \boldsymbol{V}^l, \boldsymbol{Y}^l, y^l) - L_\rho(\boldsymbol{\Theta}^{l+1}, w^{l+1}, \boldsymbol{V}^{l+1}, \boldsymbol{Y}^{l+1}, y^{l+1}) \\
\geq &\frac{\rho}{2} \left\| \mathfrak{d} w^{l+1} - \mathfrak{d} w^l \right\|_2^2 + \frac{\rho}{2} \left\| \mathcal{L} w^{l+1} - \mathcal{L} w^l \right\|_2^2 - \rho \left\| \mathfrak{d} w^{l+1} - d \right\|_2^2 - \rho \left\| \mathcal{L} w^{l+1} - \boldsymbol{\Theta}^{l+1} \right\|_{\mathrm{F}}^2 \\
&+ r \left( \mathcal{L} w^l \right) - r \left( \mathcal{L} w^{l+1} \right) - \left\langle \nabla r \left( \mathcal{L} w^{l+1} \right), \mathcal{L} w^l - \mathcal{L} w^{l+1} \right\rangle \\
\geq &\frac{\rho}{2} \left\| \mathfrak{d} w^{l+1} - \mathfrak{d} w^l \right\|_2^2 + \frac{\rho - L_r}{2} \left\| \mathcal{L} w^{l+1} - \mathcal{L} w^l \right\|_{\mathrm{F}}^2 - \frac{1}{\rho} \left\| y^{l+1} - y^l \right\|_2^2 - \frac{1}{\rho} \left\| \boldsymbol{Y}^{l+1} - \boldsymbol{Y}^l \right\|_{\mathrm{F}}^2,
\end{aligned}
\tag{69}
$$

where the last inequality is due to the fact that $r(\boldsymbol{L})$ is a concave function and has $L_r$-Lipschitz continuous gradient where $L_r > 0$ is a constant, and thus we obtain

$$r \left( \mathcal{L} w^l \right) - r \left( \mathcal{L} w^{l+1} \right) - \left\langle \nabla r \left( \mathcal{L} w^{l+1} \right), \mathcal{L} w^l - \mathcal{L} w^{l+1} \right\rangle \geq -\frac{L_r}{2} \left\| \mathcal{L} w^{l+1} - \mathcal{L} w^l \right\|_{\mathrm{F}}^2. \tag{70}$$

By calculation, we obtain that if $\rho$ is sufficiently large such that

$$\rho \geq \max \left( 2L_r, \ \max_l \frac{C \left( \left\| \boldsymbol{Y}^{l+1} - \boldsymbol{Y}^l \right\|_{\mathrm{F}}^2 + \left\| y^{l+1} - y^l \right\|_2^2 \right)^{\frac{1}{2}}}{\left( \left\| \mathcal{L} w^{l+1} - \mathcal{L} w^l \right\|_{\mathrm{F}}^2 + \left\| \mathfrak{d} w^{l+1} - \mathfrak{d} w^l \right\|_2^2 \right)^{\frac{1}{2}}} \right), \tag{71}$$

holds with some constant $C > 2$, then, together with (57), we obtain that

$$L_\rho(\boldsymbol{\Theta}^l, w^l, \boldsymbol{V}^l, \boldsymbol{Y}^l, y^l) \geq L_\rho(\boldsymbol{\Theta}^{l+1}, w^l, \boldsymbol{V}^l, \boldsymbol{Y}^l, y^l) \geq L_\rho(\boldsymbol{\Theta}^{l+1}, w^{l+1}, \boldsymbol{V}^{l+1}, \boldsymbol{Y}^{l+1}, y^{l+1}), \tag{72}$$

for any $l \in \mathbb{N}_+$.

∎

Now we are ready to prove Theorem 4. By Lemma 12, the sequence $\left\{\left(\boldsymbol{\Theta}^l, \boldsymbol{w}^l, \boldsymbol{V}^l, \boldsymbol{Y}^l, \boldsymbol{y}^l\right)\right\}$ is bounded. Therefore, there exists at least one convergent subsequence $\left\{\left(\boldsymbol{\Theta}^{l_s}, \boldsymbol{w}^{l_s}, \boldsymbol{V}^{l_s}, \boldsymbol{Y}^{l_s}, \boldsymbol{y}^{l_s}\right)\right\}_{s\in\mathbb{N}}$, which converges to a limit point denoted by $\left\{\left(\boldsymbol{\Theta}^{l\infty}, \boldsymbol{w}^{l\infty}, \boldsymbol{V}^{l\infty}, \boldsymbol{Y}^{l\infty}, \boldsymbol{y}^{l\infty}\right)\right\}$. By Lemma 13, we obtain that $L_\rho\left(\boldsymbol{\Theta}^l, \boldsymbol{w}^l, \boldsymbol{V}^l, \boldsymbol{Y}^l, \boldsymbol{y}^l\right)$ is monotonically decreasing and lower bounded, and thus is convergent. Note that the function $\log\det^*(\boldsymbol{\Theta})$ is continuous over the set $\mathcal{S} = \left\{\boldsymbol{\Theta} \in \mathcal{S}_+^p | \operatorname{rank}(\boldsymbol{\Theta}) = p - k\right\}$.

We can then obtain

$$\lim_{l\to+\infty} L_\rho\left(\boldsymbol{\Theta}^l, \boldsymbol{w}^l, \boldsymbol{V}^l, \boldsymbol{Y}^l, \boldsymbol{y}^l\right) = L_\rho\left(\boldsymbol{\Theta}^\infty, \boldsymbol{w}^\infty, \boldsymbol{V}^\infty, \boldsymbol{Y}^\infty, \boldsymbol{y}^\infty\right) = L_\rho\left(\boldsymbol{\Theta}^{l\infty}, \boldsymbol{w}^{l\infty}, \boldsymbol{V}^{l\infty}, \boldsymbol{Y}^{l\infty}, \boldsymbol{y}^{l\infty}\right).$$

Then, (69), (71) and (72) together yields

$$
\begin{aligned}
L_\rho(\boldsymbol{\Theta}^l, \boldsymbol{w}^l, \boldsymbol{V}^l, \boldsymbol{Y}^l, \boldsymbol{y}^l) &- L_\rho(\boldsymbol{\Theta}^{l+1}, \boldsymbol{w}^{l+1}, \boldsymbol{V}^{l+1}, \boldsymbol{Y}^{l+1}, \boldsymbol{y}^{l+1}) \\
&\geq \frac{C^2 - 4}{4}\rho\left(\left\|\mathcal{L}\boldsymbol{w}^{l+1} - \boldsymbol{\Theta}^{l+1}\right\|_F^2 + \left\|\mathfrak{d}\boldsymbol{w}^{l+1} - \boldsymbol{d}\right\|_2^2\right).
\end{aligned}
\tag{73}
$$

Thus, we obtain

$$\lim_{l\to+\infty}\left\|\mathcal{L}\boldsymbol{w}^l - \boldsymbol{\Theta}^l\right\|_F = 0, \quad \text{and} \quad \lim_{l\to+\infty}\left\|\mathfrak{d}\boldsymbol{w}^l - \boldsymbol{d}\right\|_2 = 0. \tag{74}$$

Obviously, $\left\|\mathcal{L}\boldsymbol{w}^{l_s} - \boldsymbol{\Theta}^{l_s}\right\|_F \to 0$ and $\left\|\mathfrak{d}\boldsymbol{w}^{l_s} - \boldsymbol{d}\right\|_2 \to 0$ also hold for any subsequence as $s \to +\infty$, which implies that $\boldsymbol{Y}^{l\infty}$ and $\boldsymbol{y}^{l\infty}$ satisfy the condition of stationary point of $L_\rho(\boldsymbol{\Theta}, \boldsymbol{w}, \boldsymbol{V}, \boldsymbol{Y}, \boldsymbol{y})$ with respect to $\boldsymbol{Y}$ and $\boldsymbol{y}$, respectively. By (64) and (62), we also have

$$\lim_{l\to+\infty}\left\|\boldsymbol{Y}^{l+1} - \boldsymbol{Y}^l\right\|_F = 0, \quad \text{and} \quad \lim_{l\to+\infty}\left\|\boldsymbol{y}^{l+1} - \boldsymbol{y}^l\right\|_2 = 0. \tag{75}$$

Together with (69), we obtain

$$\lim_{l\to+\infty}\left\|\mathfrak{d}\boldsymbol{w}^{l+1} - \mathfrak{d}\boldsymbol{w}^l\right\|_2 = 0 \quad \text{and} \quad \lim_{l\to+\infty}\left\|\mathcal{L}\boldsymbol{w}^{l+1} - \mathcal{L}\boldsymbol{w}^l\right\|_F = 0. \tag{76}$$

Recall that $\boldsymbol{V}^l$ contains the $k$ eigenvectors associated with the $k$ smallest eigenvalues of $\mathcal{L}\boldsymbol{w}^l$, and thus we can check that

$$\lim_{l\to+\infty}\left\|\boldsymbol{V}^{l+1} - \boldsymbol{V}^l\right\|_F = 0. \tag{77}$$

For the limit point $\left\{\left(\boldsymbol{\Theta}^{l\infty}, \boldsymbol{w}^{l\infty}, \boldsymbol{V}^{l\infty}, \boldsymbol{Y}^{l\infty}, \boldsymbol{y}^{l\infty}\right)\right\}$ of any subsequence $\left\{\left(\boldsymbol{\Theta}^{l_s}, \boldsymbol{w}^{l_s}, \boldsymbol{V}^{l_s}, \boldsymbol{Y}^{l_s}, \boldsymbol{y}^{l_s}\right)\right\}_{s\in\mathbb{N}}$, $\boldsymbol{\Theta}^{l\infty}$ minimizes the following subproblem

$$
\begin{aligned}
\boldsymbol{\Theta}^{l\infty} &= \underset{\substack{\operatorname{rank}(\boldsymbol{\Theta})=p-k \\ \boldsymbol{\Theta}\succeq\boldsymbol{0}}}{\arg\min} \; -\log\det^*(\boldsymbol{\Theta}) + \left\langle\boldsymbol{\Theta}, \boldsymbol{Y}^{l\infty-1}\right\rangle + \frac{\rho}{2}\left\|\boldsymbol{\Theta} - \mathcal{L}\boldsymbol{w}^{l\infty-1}\right\|_F^2 \\
&= \underset{\substack{\operatorname{rank}(\boldsymbol{\Theta})=p-k \\ \boldsymbol{\Theta}\succeq\boldsymbol{0}}}{\arg\min} \; -\log\det^*(\boldsymbol{\Theta}) + \left\langle\boldsymbol{\Theta}, \boldsymbol{Y}^{l\infty}\right\rangle + \frac{\rho}{2}\left\|\boldsymbol{\Theta} - \mathcal{L}\boldsymbol{w}^{l\infty}\right\|_F^2 \\
&\qquad\qquad - \left\langle\boldsymbol{\Theta}, \boldsymbol{Y}^{l\infty} - \boldsymbol{Y}^{l\infty-1}\right\rangle + \rho\left\langle\boldsymbol{\Theta}, \mathcal{L}\boldsymbol{w}^{l\infty} - \mathcal{L}\boldsymbol{w}^{l\infty-1}\right\rangle.
\end{aligned}
$$

By (75) and (76), we conclude that $\boldsymbol{\Theta}^{l\infty}$ satisfies the condition of stationary point of $L_\rho(\boldsymbol{\Theta}, \boldsymbol{w}, \boldsymbol{V}, \boldsymbol{Y}, \boldsymbol{y})$ with respect to $\boldsymbol{\Theta}$. Similarly, $\boldsymbol{w}^{l\infty}$ minimizes the subproblem

$$
\begin{aligned}
\boldsymbol{w}^{l\infty} &= \underset{\boldsymbol{w}\geq\boldsymbol{0}}{\arg\min} \; r(\mathcal{L}\boldsymbol{w}) + \frac{\rho}{2}\boldsymbol{w}^\top\left(\mathfrak{d}^*\mathfrak{d} + \mathcal{L}^*\mathcal{L}\right)\boldsymbol{w} + \left\langle\boldsymbol{w}, \mathfrak{d}^*\left(\boldsymbol{y}^{l\infty-1} - \rho\boldsymbol{d}\right)\right\rangle \\
&\qquad + \left\langle\boldsymbol{w}, \mathcal{L}^*\left(\eta\boldsymbol{V}^{l\infty-1}\left(\boldsymbol{V}^{l\infty-1}\right)^\top - \boldsymbol{Y}^{l\infty-1} - \rho\boldsymbol{\Theta}^{l\infty}\right)\right\rangle \\
&= \underset{\boldsymbol{w}\geq\boldsymbol{0}}{\arg\min} \; r(\mathcal{L}\boldsymbol{w}) + \frac{\rho}{2}\boldsymbol{w}^\top\left(\mathfrak{d}^*\mathfrak{d} + \mathcal{L}^*\mathcal{L}\right)\boldsymbol{w} + \left\langle\boldsymbol{w}, \mathfrak{d}^*\left(\boldsymbol{y}^{l\infty} - \rho\boldsymbol{d}\right)\right\rangle \\
&\qquad + \left\langle\boldsymbol{w}, \mathcal{L}^*\left(\eta\boldsymbol{V}^{l\infty}\left(\boldsymbol{V}^{l\infty}\right)^\top - \boldsymbol{Y}^{l\infty} - \rho\boldsymbol{\Theta}^{l\infty}\right)\right\rangle + \left\langle\mathcal{L}\boldsymbol{w}, \boldsymbol{Y}^{l\infty} - \boldsymbol{Y}^{l\infty-1}\right\rangle \\
&\qquad + \left\langle\mathfrak{d}\boldsymbol{w}, \boldsymbol{y}^{l\infty-1} - \boldsymbol{y}^{l\infty}\right\rangle + \eta\left\langle\mathcal{L}\boldsymbol{w}, \boldsymbol{V}^{l\infty-1}\left(\boldsymbol{V}^{l\infty-1}\right)^\top - \boldsymbol{V}^{l\infty}\left(\boldsymbol{V}^{l\infty}\right)^\top\right\rangle.
\end{aligned}
$$

By (75), (76) and (77), $\boldsymbol{w}^{l\infty}$ satisfies the condition of stationary point of $L_\rho(\boldsymbol{\Theta}, \boldsymbol{w}, \boldsymbol{V}, \boldsymbol{Y}, \boldsymbol{y})$ with respect to $\boldsymbol{w}$. $\boldsymbol{V}^\infty$ minimizes the subproblem

$$\boldsymbol{V}^{l\infty} = \underset{\boldsymbol{V}\in\mathbb{R}^{p\times k}}{\arg\min} \operatorname{tr}\left(\boldsymbol{V}^\top \mathcal{L} \boldsymbol{w}^{l\infty} \boldsymbol{V}\right), \quad \text{subject to } \boldsymbol{V}^\top \boldsymbol{V} = \boldsymbol{I},$$

which implies that $\boldsymbol{V}^{l\infty}$ satisfies the condition of stationary point of $L_\rho(\boldsymbol{\Theta}, \boldsymbol{w}, \boldsymbol{V}, \boldsymbol{Y}, \boldsymbol{y})$ with respect to $\boldsymbol{V}$. To sum up, we can conclude that any limit point $\left\{\left(\boldsymbol{\Theta}^{l\infty}, \boldsymbol{w}^{l\infty}, \boldsymbol{V}^{l\infty}, \boldsymbol{Y}^{l\infty}, \boldsymbol{y}^{l\infty}\right)\right\}$ of the sequence generated by Algorithm 2 is a stationary point of $L_\rho(\boldsymbol{\Theta}, \boldsymbol{w}, \boldsymbol{V}, \boldsymbol{Y}, \boldsymbol{y})$. ∎