# OpenReview forum: "Graphical Models in Heavy-Tailed Markets"
_NeurIPS.cc/2021/Conference — NeurIPS 2021 Poster_

### Official Review · Reviewer_M9op · 2021-07-14

**Rating:** 7
**Confidence:** 5

**Summary:**

The paper proposes a framework for learning a graphical model when the data generating distribution is heavy-tailed. More specifically, the author(s) propose a graph learning estimator under Markov Random Field framework when the data is generated from a Student's t-distribution. A scalable algorithm is proposed to handle connected and K-component graphs. The methodology is particularly applicable to financial time series data. The algorithm is shown to be superior in performance compared to existing methods in both simulated and real data applications.

**Ethical Concerns:**

None.

**Limitations And Societal Impact:**

I didn't see any negative societal impact of their work.

**Main Review:**

The paper proposes a novel algorithm based on the alternating direction method of multipliers (ADMM) to a pertinent problem in the graphical model viz. estimating graph structure in the heavy-tailed distribution setting. The problem has a number of applications in financial time series especially clustering the financial time series. The algorithm looks technically correct and the numerical performances are good. However, I have following queries for the author(s) mainly centred on the framework and the methodology

-I understand that the lack of constraints on the degree of the nodes may lead to graphs with isolated nodes however, I am not sure whether in real world applications one will always have the number d available. The information on the graph whether it is sparse or dense may be known but the exact number d may not be known. Any other soft constraints to consider here when d is not known?

-Both theorem 3 and theorem 4 discusses the convergence of the respective algorithms for large $\rho$. But, can author(s) shed some light about any connection with the number of iterations required to achieve the convergence?

-I am not sure whether in the K-connected components case the number k would be always known. One may has to do an estimation of K at the first stage. Am I missing something?

-Do you need any further regularization on Lw in equation (4) to achieve sparsity?

-How does one choose the rank hyperparameter $\eta$?



******** Change after author rebuttal ************

I am happy with the technical justification author(s) provided about the degree of the network and the convergence of the algorithm. Further minor technical details about the regularization and the connected components have been appropriately answered. I have now changed my score to "acceptance".


**Time Spent Reviewing:**

4 hours

---

> ### Author Response · Authors · 2021-08-09
> **Response to Reviewer M9op**
>
> Thank you very much for your constructive review.  Please find below our responses to
> your concerns. We hope we addressed them appropriately and would appreciate if you could
> reevaluate our rating.
>
> >I understand that the lack of constraints on the degree of the nodes may lead to graphs with isolated nodes however, I am not sure whether in real world applications one will always have the number d available. The information on the graph whether it is sparse or dense may be known but the exact number d may not be known. Any other soft constraints to consider here when d is not known?
>
> - As mentioned by the reviewer, and as we emphasize in the manuscript, constraining the degrees of the
> nodes is paramount to avoid graphs with isolated nodes. With that requirement in mind, there
> are basically three approaches that can be taken: (1) Assume the weighted degrees to be uniform
> (that is the approach we chose in our manuscript, for reasons which will be discussed in more details below); (2) Estimate the degrees prior to estimating the whole network, e.g., as the diagonal of the
> inverse of the sample covariance matrix; or (3) consider the degrees as an optimization variable
> and use soft-constraints such as $\sum(log(d_i))$ to avoid any $d_i$ of being zero.
> We chose approach (1) primarily because we are more interested in the relative weight between
> node connections, i.e., the value of each weighted-edge relative to a total degree of 1. Note that
> this approach does not necessarily limits the integer number of connections that one node may have,
> which is often a qualitative measure for the importance of a node in a network. Having said that,
> one could consider using the soft-constraint $\sum(log(d_i))$ if one is interested in measuring the
> weighted-degree. This should be straightforward to accommodate in our framework.
>
> > Both theorem 3 and theorem 4 discusses the convergence of the respective algorithms for large $\rho$. But, can author(s) shed some light about any connection with the number of iterations required to achieve the convergence?
>
> * The first algorithm in the paper is expected to converge to an $\epsilon$-stationary point with the $O(1/\epsilon^2)$ iteration complexity, following from Theorem 3.12 of [1]. However, more efforts are needed to formally establish this  convergence rate for the first algorithm, because it has to handle two equality constraints instead of one in [1].
> The second algorithm is still expected to enjoy the $O(1/\epsilon^2)$ iteration complexity to reach an $\epsilon$-stationary point, according to the results in [2] which reported the convergence rate of ADMM in solving the problem over Riemannian manifold. Note that the second algorithm solves the problem (16) with the rank constraint and orthogonality constraint, where the set of rank constraint is a submanifold embedded in Euclidean spaces, and the set of orthogonality constraint is a Stiefel manifold.  However, beyond [2], more efforts are still needed to establish this iteration complexity for the second algorithm, since it solves the problem (16) with two equality constraints instead of one in [2].
>
> [1] Bo Jiang, Tianyi Lin, Shiqian Ma, and Shuzhong Zhang. “Structured nonconvex and nonsmooth optimization: algorithms and iteration complexity analysis.” Computational Optimization and Applications, 72:115-157, 2019.
>
> [2] Junyu Zhang, Shiqian Ma, and Shuzhong Zhang. “Primal-dual optimization algorithms over Riemannian manifolds: an iteration complexity analysis.” Mathematical Programming, 184:1-46, 2019.
>
> > I am not sure whether in the K-connected components case the number k would be always known. One may has to do an estimation of K at the first stage. Am I missing something?
>
> * The value of $k$ is assumed to be known throughout our manuscript. In practice,
> whether that is true or not will often depend of the application at hand.
> In our case, products in financial markets are often structured and divided in categories,
> which makes it clear to infer the value of $k$ from the application itself. In case there
> is no clear value of $k$ from the application, we envision that one could split the dataset
> in training and testing datasets and compute the out-of-sample log-likelihood for different
> values of $k$ in a way to find the best value of $k$ that explains the testing dataset.
>
> > Do you need any further regularization on Lw in equation (4) to achieve sparsity?
>
> * We do not consider sparse-promoting regularization functions, as we observed that graphs
> learned from the Student-t log-likelihood with Laplacian constraints already present
> a satisfactory level of sparsity. Nonetheless, sparsity regularizations may be easily integrated
> into (4) without much effort. For instance, one could consider including the logarithm approximation
> to the $\ell_0$-norm (see e.g., [3]), which could be easily handled in the Majorization-Minimization step of our
> framework.
>
> [3] Candes (2007) Enhancing Sparsity by Reweighted L1 Minimization. https://arxiv.org/pdf/0711.1612
>
> > How does one choose the rank hyperparameter $\eta$?
>
> * The rank hyperparameter is easily handled by an adaptive update similar to those employed to
> update the ADMM hyperparameter. More precisely, the algorithm starts with a small value for $\eta$,
> and after each iteration the algorithm checks whether the rank constraint in $Lw$ is met, if not,
> then $\eta$ is increased by a factor of $2$. We observe in practice that this simple heuristic is very
> effective.

---

### Official Review · Reviewer_C54j · 2021-07-16

**Rating:** 7
**Confidence:** 4

**Summary:**

The paper proposes a new estimator for learning graphical models under the t-distribution assumption for data generation process. For the new estimator, an ADMM-based algorithm is designed to solve the associated optimization problem in order to find the graph structures. The new estimator is evaluated extensively on real-world datasets to demonstrate its effectiveness. The paper also establishes the convergence of the proposed algorithm.


**Limitations And Societal Impact:**

Yes.

**Main Review:**

Originality:

The proposed estimator is not entirely new in the literature, given that t-distribution has been used in graphical models before (see [1]). Also, it is a direct result of well-known maximum-likelihood principle, so deriving such estimator should not be difficult. For the ADMM-type learning algorithm, I acknowledge that there are numbers of technical details to be figured out in order to have the exact algorithm, but at the high level this seems to be a standard application of ADMM recipe. So it seems to me that the novelty of this paper is somewhat limited.


Quality:

In terms of technical quality, though the ADMM algorithm seems straightforward, it is nice to see that the paper proves its convergence to stationary points, which requires some efforts and skills. Regarding the algorithm, one thing that is not clear to me is the reason why the trace regularization term is needed in Eq. (16). Conceptually if we drop the regularization term, the new formulation is still equivalent to problem (16) (in the sense that they would end up with the same set of minimizers if globally optimality could be achieved), as the constraints $rank(\Theta) = p - k$ and $\Theta = Lw$ are imposed. I understand that due to the nonconvexity, ADMM algorithm could end up with different solutions for the two formulations (with or without the regularization term), but it is not obvious to me that the regularized one could work better or there is a need to introduce it. This regularization seems to add some novelty to this paper, so I hope that either theoretical or empirical comparison could be provided to support it (e.g., faster convergence or better quality of solutions), otherwise the additional computational cost seems not well-justified.

For the practical perspective, I like the extensive evaluation of the proposed algorithm on different datasets, showing its usefulness and advantage, but it would be better to extend the comparison beyond the algorithms that just work for Gaussian distributions. Below are a few questions I have for the experiments.

1. For S&P500 stocks, is there a particular reason for choosing the three GICS sectors (Communication Services, Utilities and Real Estate), as opposed to other sectors?
2. Have you considered using some type of residual returns from econometrics literature instead of log returns? If you use GICS sectors as the ground truth for clustering, residual returns may be preferred, as sometimes the co-movement of stock returns is mainly driven by some econometrics factors, which may connect stocks across different sectors.


Clarity:

Overall the presentation of the paper is clear. The description of the problem setup and the derivation of the algorithm are easy to follow. The figures shown in the experiments give an nice qualitatively demonstration of the superiority of the algorithm.

Significance:

The proposed algorithm seems to be readily applicable to a few real-world problems, as demonstrated in the experiment section. So it should be a good addition to this area for handling heavy-tailed distributions.

[1] Finegold, Michael A., and Mathias Drton. "Robust graphical modeling with t-distributions." In UAI 09, pp. 169-176. 2009.



********** post-rebuttal changes **********

After reading author response (especially on the concern about algorithmic novelty), I would like to increase my rating from 6 to 7.

**Time Spent Reviewing:**

5

---

> ### Author Response · Authors · 2021-08-09
> **Response to Reviewer C54j**
>
> Thank you very much for your constructive review.  Please find below our responses to
> your concerns. We hope we addressed them appropriately and would appreciate if you could
> reevaluate our rating.
>
> > Regarding the algorithm, one thing that is not clear to me is the reason why the trace regularization term is needed in Eq. (16).
>
> * In practice we observed that the formulation with the regularization not only returns
> better solutions, but it also converges faster. The additional computational cost of having
> such regularization is then justified. On an intuitive level, by having the
> regularization term we are encouraging the projected gradient descent update of $w$
> to go towards directions that make $\mathcal{L}w$ satisfy $rank(\mathcal{L}w) = p - k$.
>
> > For S&P500 stocks, is there a particular reason for choosing the three GICS sectors (Communication Services, Utilities and Real Estate), as opposed to other sectors?
>
> * There is no particular reason for this selection of sectors. In fact, in practice we have conducted
> experiments for different number of sectors, which revealed similar results.
>
> > Have you considered using some type of residual returns from econometrics literature instead of log returns? If you use GICS sectors as the ground truth for clustering, residual returns may be preferred, as sometimes the co-movement of stock returns is mainly driven by some econometrics factors, which may connect stocks across different sectors.
>
> * Yes, that is a very interesting question that we have investigated before.
> In particular, we have considered modeling the returns matrix as a principal component analisys (PCA)
> factor model and then using the residual returns to learn the graph structures.
> Interestingly, we observe that, as long as we scale each stock timeseries by its volatility,
> the estimated graphs are nearly the same regardless of whether returns or
> residual returns were used as input. There are two reasons that leads to that: 1) observe that our
> estimator depends on the data only through the quadratic forms $x_i^\top Lw x_i$, and since
> $Lw 1 = 0$, the component of $x_i$ that is parallel to the all-one vector does not affect the
> estimation; 2) consider the sample covariance matrix of a large number of stocks, then as a first-order
> approximation, the eigenvector associated to its largest eigenvalue (often denoted as market mode)
> is approximately constant with all of its entries having the same sign (see, e.g., [1]).
> Therefore, to first-order, our estimator is not affected by the co-movement generated by the market mode.
>
> [1] Plerou et al. (2001)
> A Random Matrix Approach to Cross-Correlations in Financial Data,
> https://arxiv.org/pdf/cond-mat/0108023.pd

---

> > ### Comment · Reviewer_C54j · 2021-08-30
> > **Algorithmic novelty**
> >
> > Thanks for the response to the comments. After reading other reviews and responses, I still have reservations on the algorithmic novelty of the paper, which will keep my original score. If you have comments about the point I made regarding novelty, could you please add them so that I can possibly re-evaluate it?

---

> > > ### Author Response · Authors · 2021-08-30
> > > **Regarding Algorithmic Novelty**
> > >
> > > We thank the reviewer for raising questions regarding the algorithmic novelty. Please find below our responses to your concerns. We hope we addressed them appropriately and would appreciate if you could reevaluate our rating.
> > >
> > > > The proposed estimator is not entirely new in the literature, given that t-distribution has been used in graphical models before (see [1]). Also, it is a direct result of well-known maximum-likelihood principle, so deriving such estimator should not be difficult. For the ADMM-type learning algorithm, I acknowledge that there are numbers of technical details to be figured out in order to have the exact algorithm, but at the high level this seems to be a standard application of ADMM recipe. So it seems to me that the novelty of this paper is somewhat limited.
> > >
> > > * While the usage of the t-distribution in graphical models have been studied in the literature before, our work applies it for the first time in the context of Laplacian-constrained graphical models. Laplacian constraints can be used to enforce many interesting properties on the estimated graphs such as k-components (which naturally leads to its application in clustering). Combined with the t-distribution, one can think of the development of applications such as robust clustering via graphs. On the flip side, those nice properties come at the expense of nonconvexities that are challenging to handle by numerical algorithms. These nonconvexities, especially in Algorithm 2, that arise in both the feasible set and the objective function, make the application of the ADMM recipe not at all standard in our case. Historically, ADMM has been well studied when the optimization problem is convex, but it is yet not widespread used in nonconvex contexts, despite huge progress made in recent years (see [1]). Hence, we needed to carefully investigate the convergence guarantees of Algorithm 1 and 2, which are presented in Theorems 3 and 4, whose proofs are in the supplementary material.
> > >
> > > [1] Wang et al 2019. Global Convergence of ADMM in Nonconvex Nonsmooth Optimization https://link.springer.com/article/10.1007/s10915-018-0757-z

---

> > > > ### Comment · Reviewer_C54j · 2021-09-02
> > > > **Post-rebuttal changes**
> > > >
> > > > Thanks for the response. Based on the points you mentioned, I would like to increase the score to 7.

---

### Official Review · Reviewer_7Jw4 · 2021-07-17

**Rating:** 8
**Confidence:** 3

**Summary:**

- This paper proposes to use the student-t distribution for data generation in financial markets, which can capture the effect of outliers better than the commonly used Gaussian distribution.

- In particular, the data is modeled using a Laplacian Markov random field with student-t distribution, and a practical ADMM based algorithm to learn the underlying graph structure is proposed.

- The algorithms are designed under two settings namely, connected graphs and k-component graphs, and convergence is proven.

- The proposed approach is shown to outperform the state-of-the-art Gaussian assumption-based approaches with real-world data from stock markets, foreign exchanges, and cryptocurrencies.



**Ethical Concerns:**

Nil

**Limitations And Societal Impact:**

 A section introducing the preliminary concepts such as MRF and the motivation to use them in financial data would be helpful.

**Main Review:**

- The paper addresses an important problem of designing graph learning algorithms under heavy-tailed data distributions which is relevant in the financial domain but challenging due to its non-convex nature.

- The convergence of proposed algorithms is analytically proven and simulations suggest better performance compared to existing algorithms.

- The paper is written well and easy to understand. The mathematical details are clear.

**Time Spent Reviewing:**

12

---

> ### Author Response · Authors · 2021-08-09
> **Response to Reviewer 7Jw4**
>
> Thank you very much for the positive review.  We appreciate your recognition of our work and we will include a paragraph in the introductory sections discussing concepts such as Markov Random Fields and their applications in financial data.

---

### Decision · Program_Chairs · 2021-09-27

**Decision:**

Accept (Poster)

**Comment:**

This paper formulates a graphical model learning problem by maximizing the likelihood of a multivariate student-t distribution subject to constraints on the Laplacian structured parameter matrix. The formulation is non convex. An explicit ADMM iteration is derived to arrive at a stationary point.

The primary strength of the paper is in its multiple experiments demonstrating the utility in the context of financial time-series of the modeling assumptions over the more commonly studied Gaussian graphical model setup.

The main weakness of the paper is that it does not contain a great deal of conceptual depth or technical novelty: the ideas, and their implementation, appear to be relatively straightforward.

Overall, the paper seems likely to have impact by spurring further work on estimation of graphical models of various types.